# Asymmetric changes of temperature in the Arctic during the Holocene based on a transient run with the CESM

Hongyue Zhang[1,2], Jesper Sjolte[2,*], Zhengyao Lu[3], Jian Liu[1,4,5,*],  Weiyi sun[1], Lingfeng Wan[6]

[1] Key Laboratory for Virtual Geographic Environment, Ministry of Education, State Key Laboratory Cultivation Base of Geographical Environment Evolution of Jiangsu Province, Jiangsu Center for Collaborative Innovation in Geographical Information Resource Development and Application, School of Geography Science, Nanjing Normal University, Nanjing 210023, China
[2] Department of Geology – Quaternary Science, Lund University, Lund, 223 62, Sweden
[3] Department of Physical Geography and Ecosystem Science, Lund University, Lund, 223 62, Sweden
[4] Jiangsu Provincial Key Laboratory for Numerical Simulation of Large-Scale Complex Systems, School of Mathematical Science, Nanjing Normal University, Nanjing 210023, China
[5] Open Studio for the Simulation of Ocean-Climate-Isotope, Qingdao National Laboratory for Marine Science and Technology, Qingdao 266237, China
[6] Institute of Advanced Ocean Study, Ocean University of China, Qingdao, China

*Correspondence to*: Jesper Sjolte (jesper.sjolte@geol.lu.se), Jian Liu (jliu@njnu.edu.cn)

**Abstract.** The Arctic temperature changes are closely linked to midlatitude weather variability and extreme events, which has attracted much attention in recent decades. Syntheses of proxy data from poleward of 60°N indicate that there was asymmetric cooling of -1.54 °C and -0.61 °C for Atlantic Arctic and Pacific Arctic during the Holocene, respectively. We also present a similar consistent cooling pattern from an accelerated transient Holocene climate simulation based on the Community Earth System Model. Our results indicate that the asymmetric Holocene Arctic cooling trend is dominated by the winter temperature variability with -0.67 °C cooling for Atlantic Arctic and 0.09 °C warming for Pacific Arctic,  which is particularly pronounced at the proxy sites. Our findings indicate that sea ice in the North Atlantic expanded significantly during the late Holocene, while a sea ice retreat is seen in the North Pacific, amplifying the cooling in the Atlantic Arctic by the sea ice feedback. The positive Arctic dipole pattern, which promotes warm southerly winds to the North Pacific, offsets parts of the cooling trend in Pacific Arctic. The Arctic dipole pattern also causes sea ice expansion in the North Atlantic, further amplifying the cooling asymmetry. We found that the temperature asymmetry is more pronounced in a simulation driven only by orbital forcing. The accelerated simulations lead to a partial delay in the feedback of climate processes. Therefore, we confirm the occurrence of the asymmetry of the Arctic temperature changes in unaccelerated simulations using ECBilt-CLIO, IPSL, and in Trace21k.

## 1 Introduction

Arctic climate is a critical component of the climate system. Since the 1990s, the changes in the Arctic climate have attracted increasing attention. Observational and model data show that the Arctic temperature variability is much greater than the global mean temperature variability, known as the Arctic amplification (AA) (Jones and Moberg, 2003; Holland and Bitz, 2003;

Serreze and Barry, 2011). The Intergovernmental Panel on Climate Change (IPCC, 2021) suggests that the Arctic temperature has likely increased more than double compared to the global average with high confidence over the last two decades. There

has been intense debate about how the AA affects the mid-latitude circulation, with some scholars suggesting that the AA increases the equator-to-pole temperature gradient and impacts the predominant westerly wind through the thermal-wind relation, which leads to disturbances in the mid-latitude circulation, further increasing the probability of extreme weather in the mid-latitudes (Smith et al., 2019; Vavrus, 2018; Screen and Simmonds, 2014; Cohen et al., 2014; Francis and Vavrus, 2012, 2015). The changes in Arctic temperature have also been directly linked to the sea ice loss or expansion, which affect

the exchanges of heat and moisture between the ocean and atmosphere as well as the salinity of the ocean (Aagaard and Carmack, 1989; Wu et al., 2004; Goosse and Fichefet, 1999; Deser et al., 2016). Changes in sea ice extent and salinity in turn leads to changes in the thermohaline circulation (Rahmstorf, 1999; Alekseev et al., 2001), which may cause significant negative impacts on the climate and maritime transportation (Ragner, 2000) in high latitude regions. All these processes have strong impacts on the regional economic development.

The different drivers of the significant increase in temperature at high latitudes over the past decades have been debated by scholars, and it is widely accepted that one of the extremely important factors is the natural variability (Polyakov and Johnson, 2000; Polyakov et al., 2002; Delworth and Knutson, 2000). For the past decades, Hoerling et al. (2001) show that due to the increase in tropical temperature, convective activity has increased, resulting in an increase in the North Atlantic Oscillation/Arctic Oscillation (NAO/AO) positive pattern, which is significantly related with the Arctic temperature (Hurrell,

1995). (Deser et al., 2015; Blackport and Kushner, 2018) attribute part of the warming of the Arctic to increasing in extratropical ocean temperature. Johannessen et al. (2004) used ECHAM4 and HadCM3 and found that the anthropogenic forcing is the dominant reason for the Arctic warming over the past decades. For the centennial and millennial time scales, the anomalies in summer insolation, driven primarily by Earth's orbital forcing, have a greater impact on the Arctic region than the low latitudes and amplify changes in temperature through positive feedbacks. Many scholars proposed a link between

Arctic temperature trends and summer insolation changes, especially during the early-mid Holocene (Park et al., 2018; Marcott et al., 2013; Kaufman et al., 2009). Arctic temperature change affects the radiative balance, which dominates the surface energy balance that controls Arctic sea ice growth and melting (Kay et al., 2008; Francis and Hunter, 2007). At the same time, sea ice changes and trends affect Arctic temperature through feedbacks and also influence Arctic atmospheric circulation changes, particularly in the lower troposphere during winter (Barnes and Screen, 2015; Overland and Wang, 2015; Francis and Skific,

2015; Cohen, 2016). The Holocene proxy data also suggest that a decrease in the Arctic sea ice relative to the present, impacting the temperature gradient between equator and pole, which might enhance warming in North American and North Pacific regions, leading to a slight decrease in temperature over East Asia, and shifting tropical rainfall northwards (Smith et al., 2019; Park et al., 2018; Hanslik et al., 2010; Funder et al., 2011; Müller et al., 2012).

From a geological perspective the characterization of the Arctic temperature variability captured by observational data is only

a small fraction of the history of climate variations. The short time period of the observed Arctic temperature change are not sufficient to represent the full range of its natural variability and to fully assess feedbacks about air-sea interactions, climate,

and ecosystems, it is still necessary to study the long-term changes, especially the millennium-scale changes, which can help shed more light on Arctic climate change in the future. For the past two millennia, temperature reconstructions (such as tree-ring, sediment and ice core records) from PAGES2k, and model data show a millennial cooling trend in the late Holocene, particularly in the Atlantic Arctic compared to the Pacific Arctic (Zhong et al., 2018). Zhong et al. (2018) presents that this cooling pattern was caused by both a weaker North Atlantic subpolar gyre and a stronger Aleutian low. The Holocene (past 11,700 years) shows millennial scale climate variations forced by changes in insolation due to orbital changes. The early-mid Holocene period is known as the Holocene thermal maximum (HTM) due to an average 5% increase in solar radiation compared to the present (Berger, 1978). The warm climate conditions were particularly pronounced at high latitudes in the early-to-middle Holocene, which is usually associated with the insolation forcing (Kaufman et al., 2004; Larsen et al., 2015; Gajewski, 2015; Briner et al., 2016; Renssen et al., 2012). Previous work examining the response to Arctic temperature change has either focused only on the trend over the whole Arctic and neglected potential regional asymmetry, or used only reconstruction data. The asymmetry of temperature change in the Arctic throughout the Holocene period, however, is evident (Fig. 1), likely associated with responses of climate modes to external climate forcings. The uncertainty and the low spatial coverage of reconstruction in temperature changes from proxy data makes it difficult to fully understand the Arctic temperature changes during the Holocene.

To overcome these limitations, a more comprehensive view of climate system changes can be obtained by using model data studies for the Holocene. There are some transient simulations covering the Holocene with different prescribed have been done during the past decades. For example, TraCE-21ka (Liu et al., 2009)) is a transient simulation with resolution of 3.75°×3.75° for exploring the climate evolution since the LGM, based on Community Climate System Model version 3(CCSM3). Bader et al., (2020) used the Max Planck Institute Earth System Model (MPI-ESM) perform transient simulations spanning the period from 8 ka BP until the 1850 to investigate the contradiction of temperature trend changes between reconstructed and model simulated. Braconnot et al., (2019) used the transient Holocene simulations based on IPSL ESM with prescribed vegetation, interactive phenology and interactive carbon cycle for exploring the climate changes coupling vegetation over the last 6000 years. In this study, the transient simulation performed by Community Earth System Model (CESM) with an acceleration factor of 10 and a recent compilation of temperature proxy data are used to investigate the characteristics of the regional temperature changes in the Arctic. Numerous studies have shown that accelerated transient simulations have the ability to study climate evolution over long time scales. For instance, Varma et al., (2012) compared the simulation results with 10 times acceleration and non-acceleration, and found that there is no significant difference in the characteristics of global surface climate change. Timm and Timmermann, (2007) used the earth system model of intermediate complexity ECBilt-CLIO to simulate the climate since the Last Glacial Maximum (LGM) by 10 times acceleration, and compared the simulation results without acceleration and found that the simulation results with 10 times acceleration reproduced well the large-scale trend of atmospheric temperature in the Holocene. Lu and Liu, (2019) found that the acceleration leads to suppressed and delayed responses mainly in the deep sea and has less robust effect on the surface and subsurface. Jing et al., (2022) compared the temperature and precipitation changes in NNU-Holocene simulation and TraCE-21ka non-acceleration simulation, and in

terms of overall trend and distribution, the temperature and precipitation patterns of NNU and TraCE-21ka are similar. We are therefore motivated to use these simulations to assess the changes in spatial pattern of Holocene Arctic temperature variability and reveal the physical mechanisms behind these changes.

In this paper, we investigate whether temperature changes differed between the Atlantic and Pacific Arctic during the Holocene, and whether changes in sea ice and sea level pressure contributed to the pattern of temperature changes. To address these questions and identify the key factors driving asymmetric temperature changes, we analyze changes in sea ice, sea level pressure, and orbital forcing from CESM NNU-Hol simulations and other transient simulations. This is critical for future studies of the climate implications of temperature changes at high latitudes. The structure of this article is as follows. Sect. 2 describes the proxy and model data used in the current study. In Sect. 3, we summarized the temperature asymmetric changes reflected in the proxy and model data in the high latitudes of the Northern Hemisphere and presented the changes in sea ice and sea level pressure during the Holocene. In Sect. 4, the characteristics of Arctic temperature changes in Holocene unaccelerated simulations are discussed. Finally, a summary is given in Sect. 5.

## 2 Method and data

### 2.1 The CESM Model and the transient simulations

This study analyzes a new transient simulation result based on the CESM, named NNU-Hol (Nanjing Normal University-Holocene), which considers more comprehensive external forcings for the Holocene climate change. The CESM is a fully-coupled, global climate model, which was launched by the National Center for Atmospheric Research (NCAR) in June 2010. It is an Earth System Model developed on the basis of CCSM. The CESM includes components for atmosphere, ocean, sea ice, as well as the land surface, and considers atmospheric chemistry, biogeochemistry, and anthropogenic forcing. It is widely used to study the mechanism of the changes in climate and environment, the interaction between natural and anthropogenic forcing for climate and for scenarios of future climate change . The CESM is coupled with several advanced modules, including the CAM5 (Community Atmosphere Model 5) used by the atmosphere module, the POP2 (Parallel Ocean Program 2) used by the ocean module, the CLM4 (Community Land Model 4) used by the land module, the CICE (The Los Alamos National Laboratory Sea-ice Model) used by the sea ice module and the CISM2.0 (The Glimmer Ice Sheet Model 2.0) used by the land ice module . The CISM2.0 is deactivated in the NNU-Hol simulation. For a more detailed introduction, please visit the official website of CESM (http://www.cesm.ucar.edu/models/cesm1.0/notable_improvements.html).

In NNU-Hol, the CESM 1.0.3 was configured to simulate the transient climate evolution of the Holocene period at a horizontal resolution of 3.75°×3.75°, forced by several external forcings (orbital parameters, solar irradiance, volcanic eruptions, greenhouse gases, and land use/land cover) accelerated by a factor of 10. With this acceleration method (Lorenz and Lohmann, 2004), climate trends and feedbacks from the past 11.95 ka BP to 1990 AD, imposed by the external forcing driven changes, are represented in the experiments with 1199 model years to save computation resources. The solar irradiance forcing comes from the reconstruction of (Vieira et al., 2011), aggregated to 10-year average timescale of solar forcing and prescribed into

the simulation. The volcanic eruption comes from the ice core based reconstructions of (Gao et al., 2022) and (Sigl et al., 2015). For the volcanic forcing, the volcanic events during the 10-year period were integrated into one volcanic eruption event. On the basis of this assumption, the horizontal diffusion of lower stratospheric aerosols was calculated using the stratospheric transport parameters. Based on the stratospheric-tropospheric folding and BD (Brewer Dobson) circulation theory latitude- and time-dependent functions to describe aerosol production and deposition (Grieser and Schonwiese, 1999; Holton et al., 1995). The greenhouse gas forcing data uses the reconstruction based on ice cores in (Joos and Spahni, 2008). The data of land use/extra-land cover forcing comes from the HYDE 3.2.1 (the History Database of the Global Environment, referred to as HYDE version 3.2.1) (Goldewijk et al., 2017) data set. The orbital parameters come from Berger (1978). The external forcing timeseries used in the NNU-Hol simulation is shown in the supplementary material. Wan et al. (2020) found that the global annual average temperature in the NNU-Hol All forcing simulation and reconstruction records from Marcott et al. (2013) has similar trends and strengths, decreasing about 0.5 K during 5.0 to 0.15 ka BP. We explore the characteristics of changes in Arctic temperature during the Holocene and try to understand its underlying mechanism based on the All forcing (AF) and orbital forcing (ORB) simulations in NNU-Hol.

There are, to our knowledge, 5 sets of climate simulations published so far covering the entire Holocene period (0-11700 ka BP), namely ECBilt-CLIO (Timm and Timmermann, 2007), FOAM (Kutzbach et al., 2008), TraCE-21ka (Liu et al., 2009), FAMOUS (Smith and Gregory, 2012) and LOVECLIM (Timmermann et al., 2014). Except for TraCE-21ka and one of the ECBilt-CLIO simulations, these simulations are accelerated by different factors. Additional unaccelerated simulations, such as the simulations based on MPI-ESM (Bader et al., 2020) and IPSL (Braconnot et al., 2019) covering only part of the Holocene period (0-8 ka BP and 0-6 ka BP respectively) have also been published in recent years. The external forcings considered in these simulations are generally a part of the combination of the Orbital Forcing (ORB), the Greenhouse Gases (GHG), the continental ice sheets (ICE), the Meltwater Flux (MWF), the Volcanic Forcing, the Landuse Forcing and Ozone Forcing.

## 2.2 Reconstructing Paleo Proxies data

In order to better understand the Holocene evolution of the earth system, a comprehensive database of paleoclimate records was compiled by Kaufman et al., (2020), which is named the Temperature 12k database.

Temperature 12k database is a global compilation of good-quality, published, temperature-sensitive proxy records (such as lake sediment, marine sediment, peat, glacier ice and pollen, etc) through the entire Holocene period. The data is mainly collected from previously published research, containing 1319 records (1162 from the Northern Hemisphere and 157 from the Southern Hemisphere), distributed in 679 sites (including 470 terrestrial and 209 marine sites) where time period cover at least 4000 years. For our study we only selected records with a resolution finer than 400 years, 15% of these records have a resolution of 50 years or finer, 39% have a resolution of 51 to 150 years, and 21% have a resolution of more than 250 years.

The data is mainly based on a collection of published reconstructions of Holocene temperature especially in the Northern Hemisphere (Routson et al., 2019; Marcott et al., 2013; Sundqvist et al., 2014; Chen et al., 2008; Wanner et al., 2011). Part of the data collected the global paleotemperature records from PAGES 2k Consortium 9 database. The compilation of Marsicek

et al., (2018) provides most pollen-based paleotemperature records and the other part comes from public repositories (such as PANGAEA and World Data Service for Paleoclimatology, NOAA). The paleotemperature records in 209 marine sites are mainly from the US-based Data Assimilation for Deep Time (DADT) project and the compilations of the German Climate Modeling Initiative (PalMod) (Jonkers et al., 2020). In this database, uncertainties are estimated by various methods. Some studies characterize uncertainty based on calibration and proxy bias, or measurement error, while others represent uncertainties after rigorous cross-validation. The uncertainties of most of these paleotemperature records from North American and European pollen and most of the marine sediment-based records (except for those from microfossil assemblages) are calculated using the Bayesian procedure in early studies (Malevich et al., 2019; Tierney and Tingley, 2014, 2018; Tierney et al., 2019). For the other proxy types, most of the paleotemperature values and their uncertainties are based on multigenerational analysis and calibration methods. Specific information about the uncertainty for each proxy recorded is available in Kaufman et al., (2020) Supplementary Table 2 as well as in the original publication. More detailed information can be found at www.ncdc.noaa.gov/paleo/study/27330.

In order to investigate the changes in temperature over the Arctic area during the Holocene, we selected records from the site, which are located above 60 degrees north latitude, and eliminated some records with shorter time series (missing more than 2500 years during the Holocene), which leaves us with 58 records.

## 2.3 Analytical and Statistical Methods

We focus on long-term temperature changes in the Arctic during the Holocene. The significance test used in this study was calculated according to the two-tailed Students t-test at the 90% (alpha = 0.1) or 95% (alpha = 0.05) confidence level. The Students t-test was used to compare the means of two groups and determine if the difference in means is statistically significant and was also used to test the statistical significance for each grid point in the figures below. The sample size of the Pacific Arctic region in the temperature proxy data is small and thus a small degree of freedom. The potential impact of temporal and spatial correlation is not taken into account in the analyses. We apply empirical orthogonal function (EOF) analysis, also known as principal component analysis (PCA), to sea level pressure changes in the Northern Hemisphere. EOF analysis is a standard analytical technique used in climate science to study patterns of spatial variability. EOF is obtained by computing the eigenvectors and eigenvalues of the spatially weighted covariance matrix of the temperature field. Equation 24 from North et al., (1982) was used as a statistical test to evaluate the separation of EOF eigenvalues (leading modes). Applying EOF to the Northern Hemisphere sea level pressure is a common method to study the Arctic dipole mode. The objective is to show the variation of the Arctic dipole mode during different periods of the Holocene (0-2 ka BP, 5-8 ka BP). As described in Section 3 below, the second mod of EOF for the Holocene 5-8 ka BP period explains 11.5% and that for 0-2 ka BP period explains 16.3% of the sea level pressure variation.

# 3 Result

## 3.1 Arctic Temperature Change

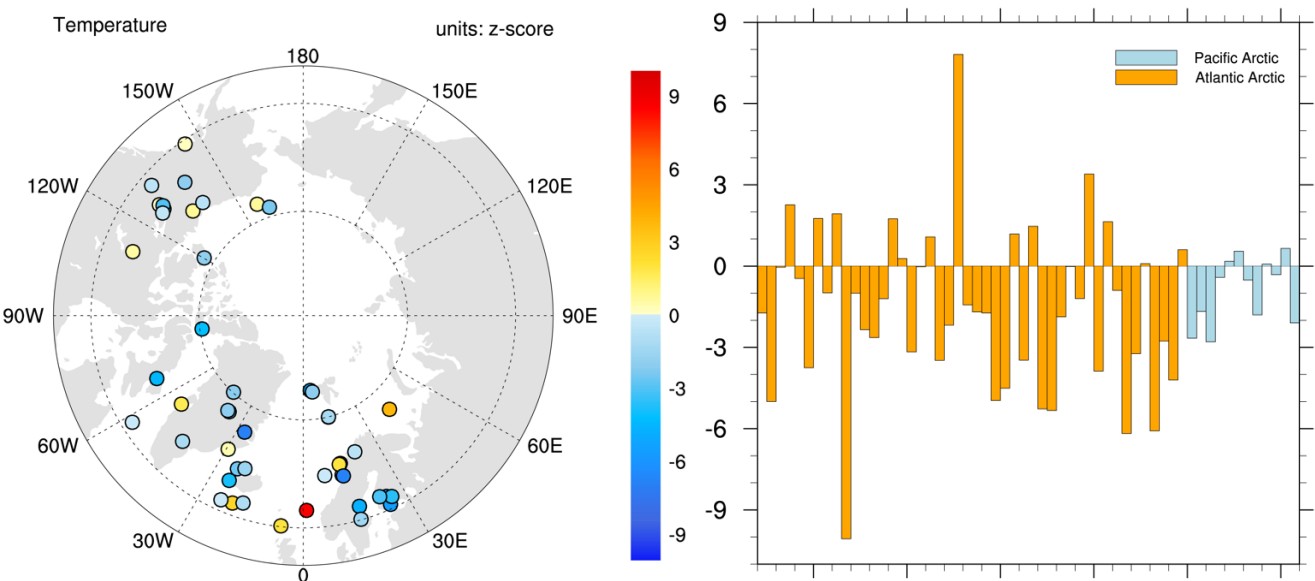

**Figure 1: (a) The annual temperature changes between two period (0-2 ka BP minus 5-8 ka BP) in Reconstructions (Temperature 12k); (b) Histogram of temperature change for each proxy site**

In order to demonstrate asymmetric temperature changes in the Arctic during the Holocene, we analyzed the Temperature 12k database (Kaufman et al., 2020). The results are shown in Fig. 1, where the circles represent the site locations of the 52 selected records. The proxy records are mainly concentrated on the Atlantic coast, Europe, Greenland, and northern Canada. The average temperature across the Arctic showed a cooling trend during the Holocene, which is consistent with previous findings (Marcott et al., 2013; Kaufman et al., 2004). To study the asymmetric changes in temperature, we divided the Arctic region into two parts, the Pacific Arctic (Lat > 60°N, Lon 90°W ~ 59°E) and the Atlantic Arctic (Lat > 60°N, Lon 60°E ~ 91°W). The proxy data shows that there is an asymmetric cooling between the two regions, with an average cooling of -1.54 °C in the Atlantic Arctic and -0.61 °C in the Pacific Arctic respectively, showing significance at the 90% confidence level ($p<0.10$) on students t-test. The degrees of freedom of the temperature proxy data in the Pacific and Arctic region is small. The histogram shows that only two proxy sites have extreme temperature variations in the Atlantic Arctic region. However, combined with the box plot of the temperature changes (Supplementary Fig. 1), the asymmetric temperature variation in the two regions is still considered robust. To further study the robustness of this asymmetric change in temperature, we likewise analyze it with model data, allowing us to also investigate the seasonal changes.

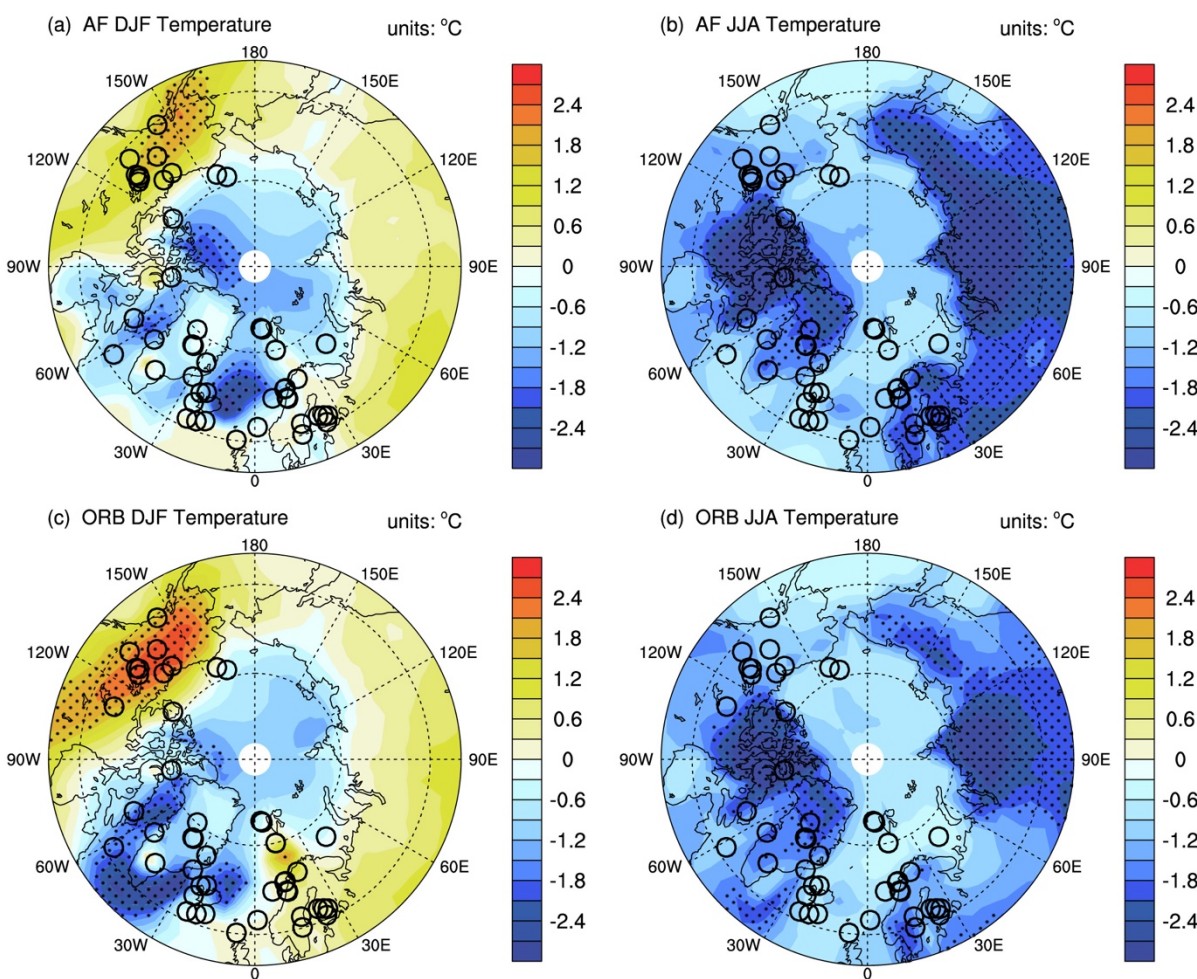


**Figure 2: (a) The DJF temperature changes between two periods (0-2 ka BP and 5-8 ka BP) in NNU-Holocene All forcing simulations (Pacific Arctic:-0.44, Atlantic Arctic:1.00, the difference is statistically significant at the 95% confidence level);(b)Same as (a) but for JJA (Pacific Arctic:3.98, Atlantic Arctic:3.81) (c) Same as (a) but for NNU-Holocene ORB forcing simulations (Pacific Arctic:-0.80, Atlantic Arctic:1.75, the difference is statistically significant at the 95% confidence level);(d)Same as (c) but for JJA (Pacific**
**Arctic:7.48, Atlantic Arctic:6.07). Areas that are dotted are significant at the 90% confidence level.**

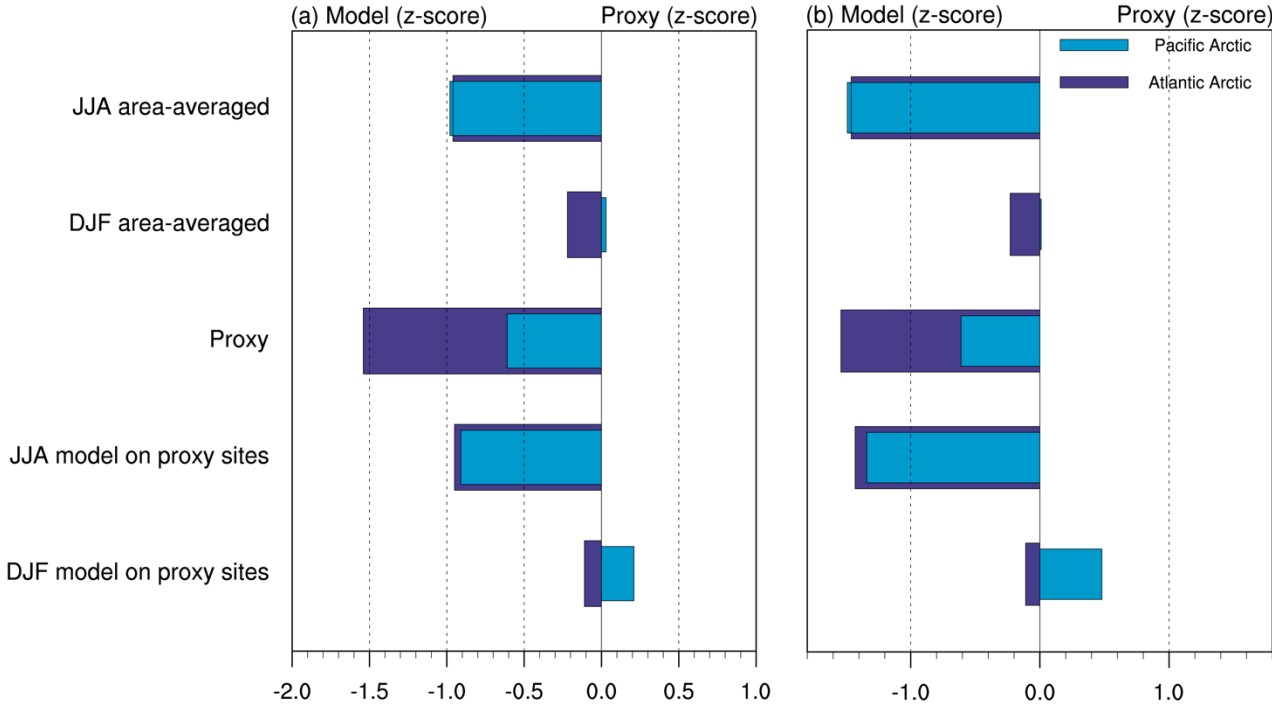

**Figure 3: 1) Model area average temperature change in JJA; 2) Model area average temperature change in DJF; 3) Annual temperature change in Proxy; 4) Model JJA average temperature change on proxy sites; 5) Model DJF average temperature change on proxy sites (a: AF simulation, b: ORB simulation);**

Here we use the model data based on the CESM, the NNU-Hol, and we choose data from All Forcing simulation (AF) and Orbital forcing simulation (ORB) in order to assess the most realistic temperature changes (AF) and the role of the orbital forcing (ORB) in the changes during the Holocene. As shown in Figure 2, to better understand the asymmetry, we further analyze the temperature changes in two seasons, DJF (December, January and February) and JJA (June, July and August). The results from AF and ORB show a similar pattern. In the JJA, the Artic exhibits a comparable cooling trend in the Atlantic

sector and the North Pacific sector, with cooling centers in Eurasia and northeastern America. However, there is significant regional asymmetric cooling in the DJF (-0.67 °C for Atlantic Arctic average cooling and 0.09 °C for Pacific Arctic average warming in AF, the difference is statistically significant at the 95% confidence level on students t-test), with the Arctic Ocean, the North Atlantic Ocean and Greenland are cooling, while the land and North Pacific Ocean are warming. Areas that are dotted are significant at the 90% confidence level in Fig. 2. Northern Canada, where numerous proxy sites are located, is the

center of the temperature warming, with the cooling center changing from the Straits in the AF to the North Atlantic in the ORB. We can see more clearly the difference of temperature asymmetry in two seasons through Figure 3, and the intensity of the asymmetry is more pronounced at the proxy sites. In the DJF of AF, the average cooling is -0.44 °C for the Atlantic Arctic proxy site and the average warming is 0.85 °C for the Pacific Arctic proxy site, while in the ORB it is -0.25 °C and 1.61°C, respectively. The difference in the mean temperature changes in the two regions is significant at the 90% confidence level in

both AF and ORB. Fig.3 shows that the temperature difference between the Pacific and Atlantic Arctic is neglectable for either the regional average or the sites station average in JJA. Therefore, the asymmetric temperature changes of the Atlantic and Pacific Arctic are dominated by the changes in DJF.

### 3.2 Sea Ice (March) and Sea Level Pressure Change

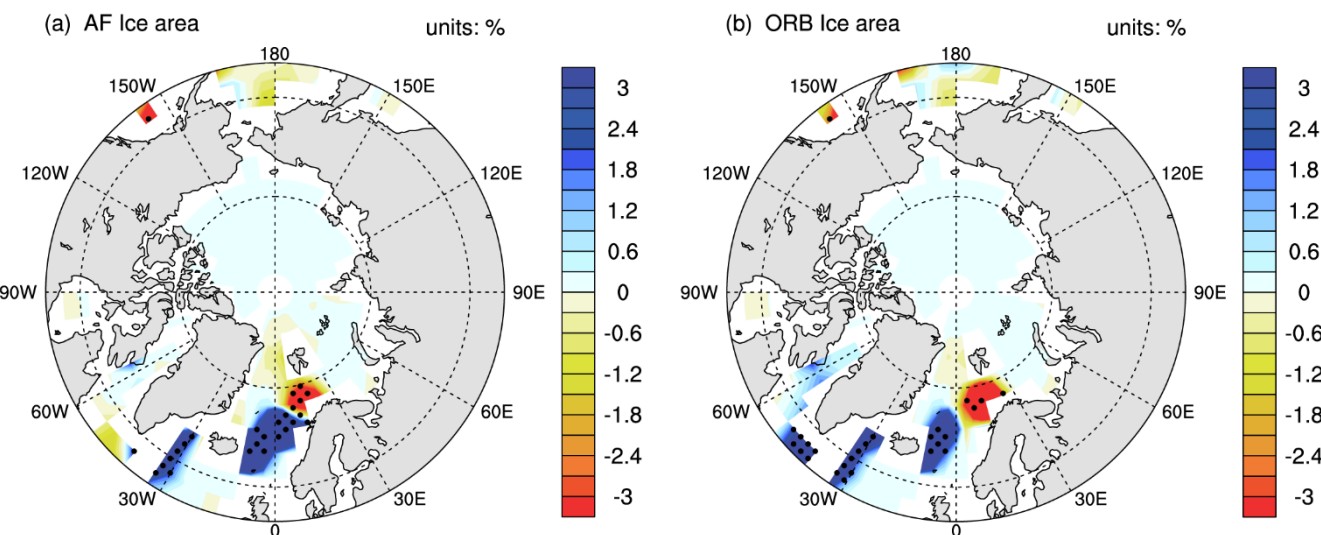

**Figure 4: (a) The March sea ice area (aice) changes between two period (0-2 ka BP and 5-8 ka BP) in NNU-Holocene All forcing simulation (regional average on Pacific Arctic:0.02% and Atlantic Arctic:0.29%); (b)Same as (a) but for ORB forcing simulation (regional average on Pacific Arctic:0.04%, Atlantic Arctic:0.29%); Areas that are dotted are significant at the 90% confidence level. The difference of the means sea ice area changes in the two regions is significant at the 90% confidence level in both AF and ORB.**

The past research has shown that sea ice is always an important factor when we discuss Arctic climate change (Jenkins and Dai, 2021). Arctic temperature changes are often tightly linked to sea ice changes, with temperature causing changes in the expansion of sea ice cover and therefore changes in surface albedo, further amplifying climate change at Arctic region (Wohlfahrt et al., 2004; Renssen, H. et al., 2005; Braconnot et al., 2007). The simulations show that the temperature asymmetry is closely related to the asymmetric change in sea-ice pattern. The difference in March sea ice area between the two periods

(0-2 ka BP and 5-8 ka BP) is fairly consistent in the AF and ORB simulations. Consistent with previous studies of Holocene sea ice concentration proxy, sea ice concentrations were low in the early-mid Holocene and increased in the mid-late Holocene during the Neoglacial (Jennings et al., 2002; de Vernal et al., 2005; Müller et al., 2012). For the Atlantic Arctic, the sea ice concentration of Chukchi Sea was not always low during the Early Holocene, while there were millennial oscillations and minimum values in the Chukchi Sea during the Neoglacial (de Vernal et al., 2005). Müller et al.,(2012)showed that for the

North Atlantic sea ice reconstruction P25 decreased significantly in the early Holocene, while in the middle Holocene 7000-3000 years BC, sea ice gradually increased. The maximum value was gradually reached in 3000-300 years. In our model, sea

ice increased throughout the Arctic during the Holocene, especially around the Norwegian Sea and Davis Strait, with the exception of the Barents Sea, where sea ice area declined significantly (significant area are dotted in Fig. 4). The large expansion of sea ice in the North Atlantic and the slight decrease in sea ice in the North Pacific leads to strong differences in surface albedo between the two regions, with more reflection of radiation in the North Atlantic and more absorption in the North Pacific, which contributes to the stronger asymmetry during winter (Fig. 3).

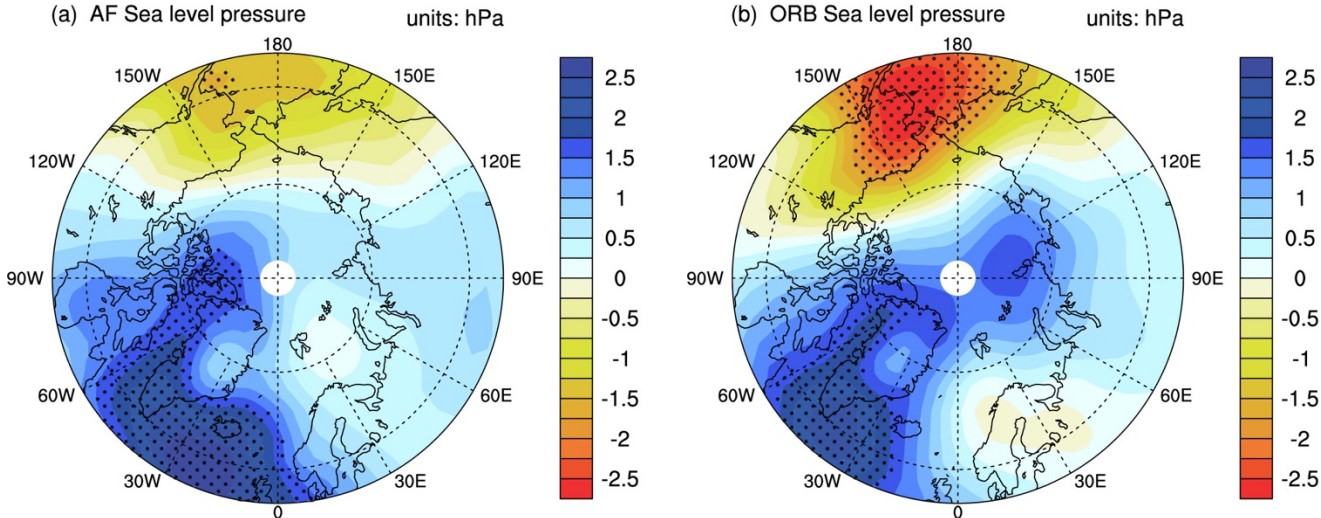

**Figure 5: (a) The DJF sea level pressure changes between two period (0-2 ka BP and 5-8 ka BP) in NNU-Holocene All forcing simulation;(b) Same as (a) but for ORB forcing simulation; Areas that are dotted are significant at the 90% confidence level.**

To understand the dynamical aspects of the stronger cooling in the North Atlantic, we investigate the regional-scale atmospheric circulation and surface wind variations related to sea-ice transport. We focus our analysis on the December–February season, because these months yield the largest impact of temperature asymmetry. In DJF, the sea level pressure (SLP) has a distinct dipole distribution. Sea level pressure increases in the North Atlantic and decreases in the North Pacific. The low pressure in the North Pacific is intensifying during the late Holocene. Based on data from the past decades, many studies (Wu et al., 2006; Niebauer et al., 1999; Stabeno et al., 2001; Rodionov et al., 2005) have demonstrated that the stronger low pressure transported warm air from the south to the North Pacific, resulting in higher temperatures in the North Pacific, offsetting parts of the Holocene cooling trend in the North Pacific and causing sea ice expansion in the North Atlantic due to atmospheric circulation. We suspect that this contributes to the regional asymmetry in Arctic temperatures. Compared with the AF results, the ORB simulation shows that the sea level pressure changes in the two regions are more contrasting and the difference is larger. This suggests that orbital forcing plays a dominant role in generating this asymmetry.

### 3.3 EOF of SLP and regressed UV wind and sea ice

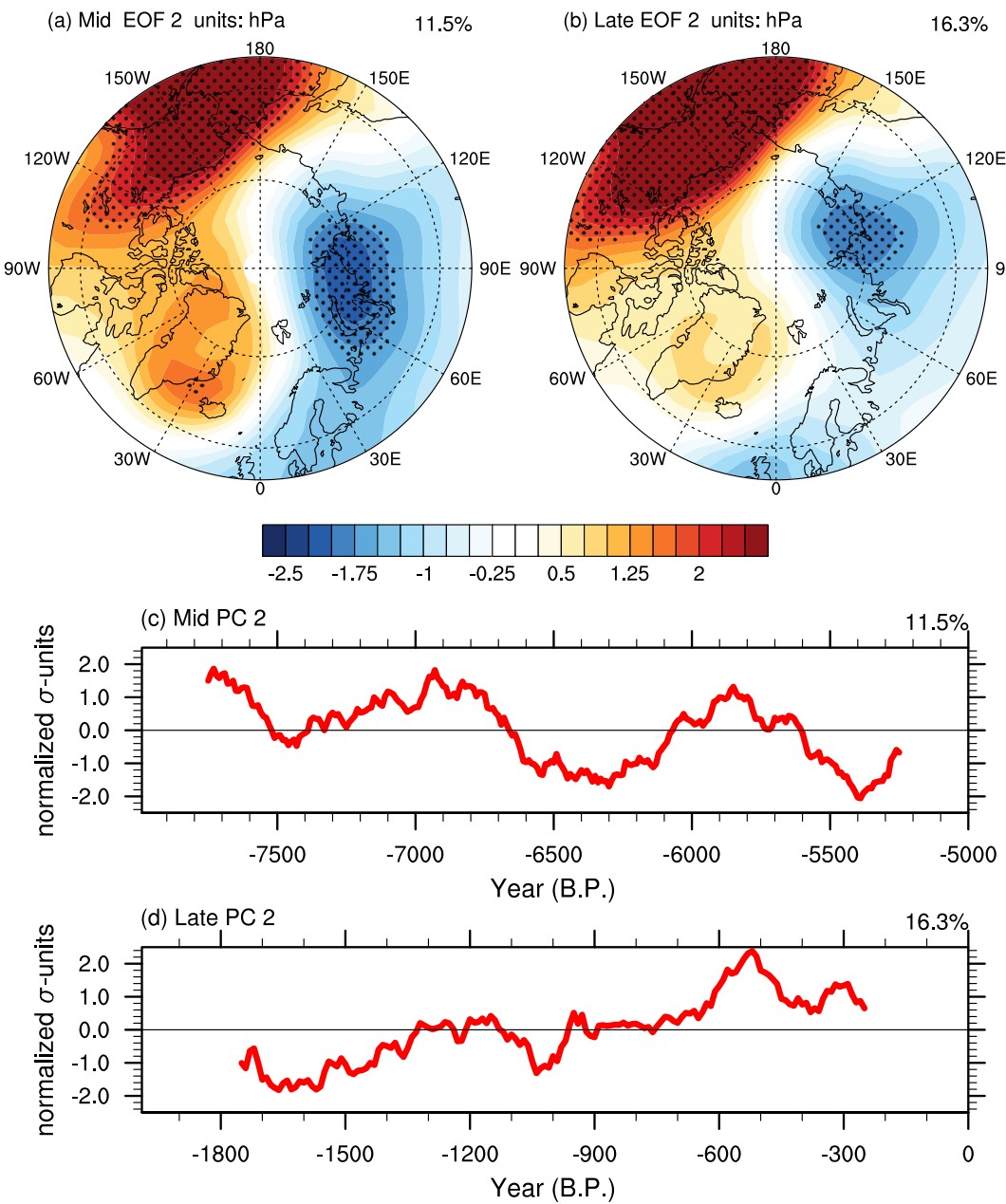

**Figure 6: The second EOF (EOF2) pattern of the Sea level pressure in NNU-Holocene All forcing simulations during 5-8 ka BP (a) and 0-2 ka BP (b); The second Principal Component time series of the EOF during 5-8 ka BP (c) and 0-2 ka BP (d) after 50yr smoothed; Areas that are dotted are significant at the 90% confidence level.**

Previous studies have established that atmospheric circulation anomalies in the Arctic atmosphere, dipole structure anomalies, are closely related to sea ice (Wu et al., 2006; Watanabe et al., 2006; Choi et al., 2019). The Arctic dipole strongly influences

sea ice movement and sea ice export. The empirical orthogonal function(EOF)method is commonly used to study the Arctic dipole in past studies (Wu et al., 2006; Skeie, 2000; Wang and Ikeda, 2000). The EOF's first leading mode of the monthly mean SLP north of 60 degrees north latitude in winter corresponds to the Arctic Oscillation pattern and the second leading mode of the EOF corresponds the of the dipole anomaly. In order to confirm our conjecture that changes in SLP and sea ice

play a role in promoting regional temperature asymmetry, we analyzed the Arctic dipole, and the results are shown in Fig. 6. The first and second leading modes are statistically independent at 95% confidence level on North significance test (North et al., 1982). In the AF simulation, the EOF pattern of the SLP is similar for both periods (0-2 ka BP and 5-8 ka BP), with the second mode showing an Arctic dipolar distribution. There are two action centers with opposite signs one center is located over the North Pacific and the other exists over the northern Eurasia (significant at the 90% confidence level), accounting for

11.5% (5-8 ka BP) and 16.3% (0-2 ka BP) of the variance. Fig. 6c, d shows that the PC2 time series of the EOF in the mid-Holocene oscillated repeatedly, while the PC2 in the late Holocene has shown a strengthening trend. Combined with the stronger SLP of the late Holocene shown in Fig. 5, it can be known that the late Holocene has a stronger Arctic Dipole pattern. Regressing on PC2 of EOF allows us to understand how the Arctic dipole pattern affects the changes in sea ice and wind, and the corresponding sea ice field and wind field can explain the physical mechanism of the Arctic dipole effect on temperature

asymmetry. To explore our conjectures, we compared two spatial patterns of sea ice and UV wind field regressions to PC2 (Fig. 7). By regressing the sea ice distribution onto the second Principal Component time series (Fig. 7), a pattern is consistent with past sea ice reconstructions (de Vernal et al., 2005; Müller et al., 2012; Jennings et al., 2002),  we show that the Arctic dipole in the late Holocene had a greater role in influencing sea ice. In the late Holocene, the Bering Strait, Chukchi seas and East Siberia were influenced by warm winds from the south, promoting strong ice melting, pushing the ice away from the

coast and increasing the temperature in these areas. On the other hand, the pressure pattern causes the Arctic Ocean sea ice to expand to the North Atlantic Ocean through Fram Strait transfer. This leads to more sea ice in the Atlantic Ocean, and therefore, more radiant heat reflected and surface temperature cooling in Atlantic Arctic due to the sea ice feedback. This pattern contributes to the asymmetry of temperature variations between the North Atlantic and the North Pacific. In the late Holocene, the wind field characteristics are more pronounced than those in the early-mid Holocene, and thus the temperature asymmetry

is more significant.

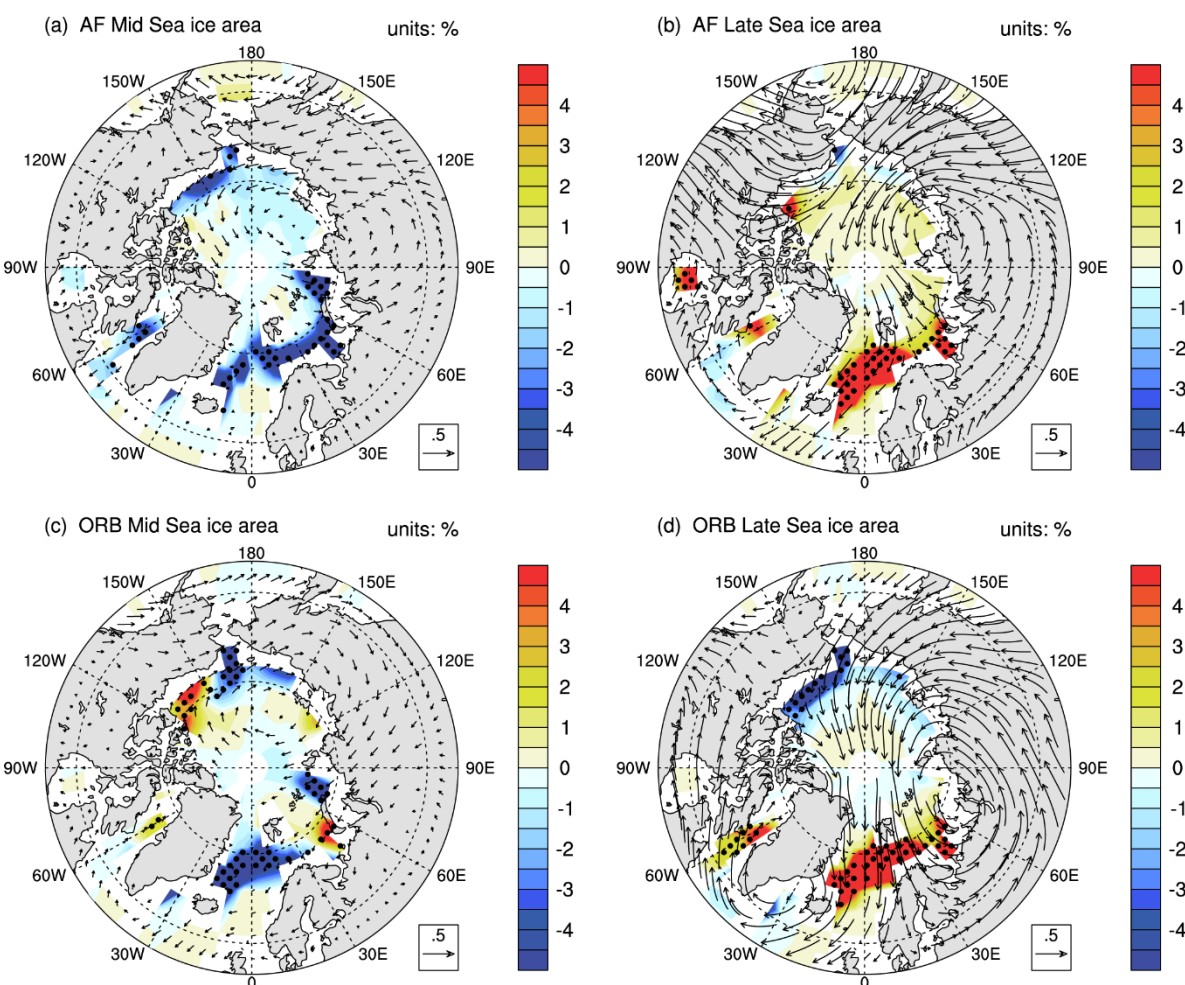

**Figure 7: The regression of UV wind and sea ice based on the second Principal Component time series of the Sea level pressure in NNU-Holocene AF simulation during 5-8 ka BP (a) and 0-2 ka BP (b); in the NNU-Holocene ORB simulation during 5-8 ka BP (c) and 0-2 ka BP (d); Areas that are dotted are significant at the 90% confidence level.**

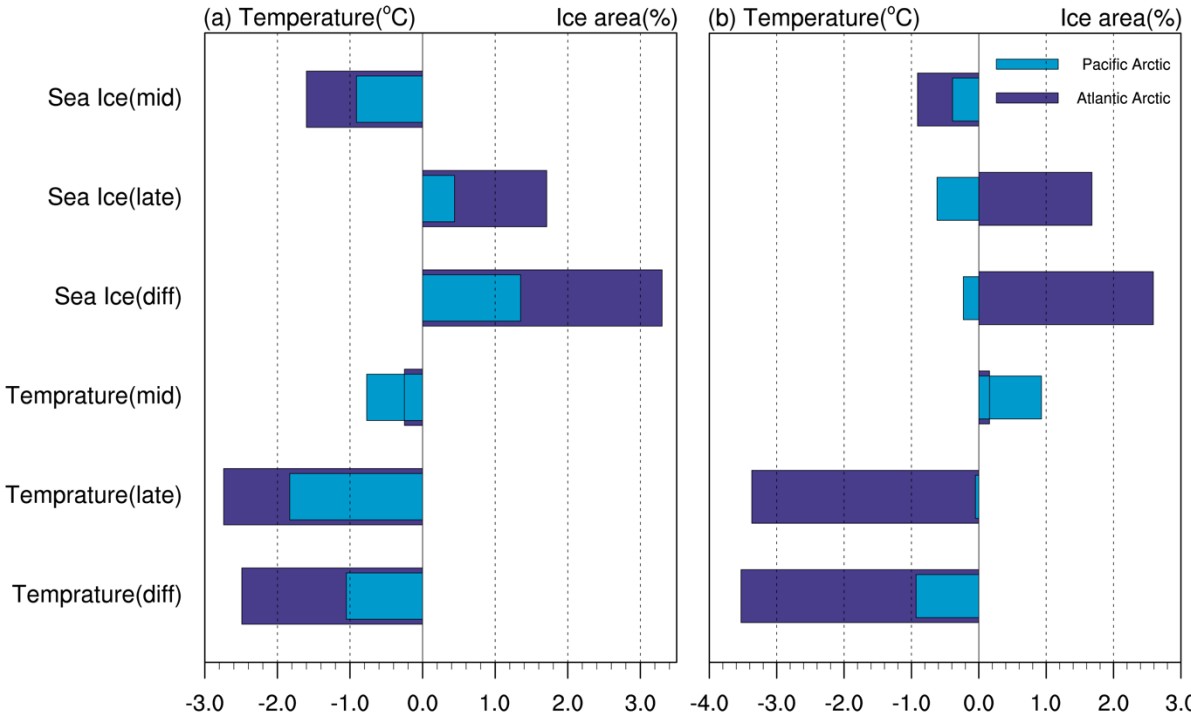


**Figure 8: 1) Average change in the regression of Sea ice on the PC2 of the SLP in early-mid Holocene; 2) same as 1) in late Holocene; 3) Differences in the regressions of Sea ice on PC2 of SLP between two period; 4) The regression of temperature average change in early-mid Holocene; 5) same as 4) in late Holocene; 6) Differences in the regressions of temperature on PC2 of SLP between two period (a: AF simulation, b: ORB simulation); the difference in sea ice and temperature between two period are statistically significant at the 90% confidence level.**


Figure 8 shows the average change in the regression of temperature and sea ice onto PC2, and the differences in sea ice and temperature between the two regions are mainly contributed by the Late Holocene period. The regression of sea ice on the PC2 of SLP increased by 1.71% in North Atlantic and 0.44% in North Pacific during the late Holocene, while decreased by -1.60% and -0.91% during the early-mid Holocene (significant at the 90% confidence level). The corresponding asymmetric cooling

in the early-mid to late Holocene was -2.74°C in the North Atlantic and -1.83°C in the North Pacific (significant at the 90% confidence level). The results of the ORB simulations in terms of asymmetry are similar to the AF simulations, but stronger. Figure 7 shows that in the ORB simulation, the late Holocene had stronger warm southerly winds and more open water in the North Pacific, while sea ice expanded more widely in the North Atlantic, resulting in stronger asymmetric temperature. The regression of temperature shows a cooling of -3.37°C and -0.05°C in the North Atlantic and North Pacific(significant at the

90% confidence level), respectively, during the late Holocene. Therefore, the regression of the UV wind, the sea ice and the temperature on PC2 show the contribution of the Arctic Dipole to the temperature asymmetry, The result indicating that it is an important factor affecting the temperature asymmetry in the Arctic.

## 4 Discussion

The results indicate that the sea level pressure and sea ice changes are an important factor for the asymmetric Arctic cooling, since these two factors show similar asymmetric variations over the North Atlantic and the North Pacific. Compared with the early-mid Holocene, the sea ice expanded in the North Atlantic during the late Holocene, which leads to more cooling in the Atlantic Arctic due to its feedback. On the other hand, the Arctic dipole pattern, which plays an important role in sea-ice expansion in the North Atlantic, promotes more cooling in Atlantic Arctic and raising the temperature in the Pacific Arctic with warm southerly winds, exacerbating the temperature asymmetry. The asymmetric pattern is consistent with the asymmetric cooling in the Arctic over the past 2,000 years (Zhong et al., 2018). (Zhong et al., 2018) suggests that the change in ocean density due to the millennial cooling has led to a slowdown of the subpolar circulation. This deceleration of the subpolar circulation reduces the heat advection in the northern North Atlantic and intensifies the cooling in Atlatntic Arctic region. Our results are qualitatively consistent with their results that sea level pressure and sea ice play an important role in the asymmetric cooling. However, it should be pointed out that meltwater forcing was not prescribed in NNU-Hol, thus the response in ocean salinity and subpolar circulation as Zhong et al. (2018) due to meltwater can be underestimated in our simulation.

It can be assumed that our accelerated simulations are largely in qualitative agreement with the unaccelerated experiments in terms of the long-term climate evolution without involving changes in the deep ocean. We present some simple results to validate our results by comparing other non-accelerated simulations. We have selected three open-access unaccelerated simulations, TraCE-21ka, ECBilt-CLIO and IPSL. The ECBilt-CLIO is an Earth System Model of Intermediate Complexity.

In summary, for the asymmetric cooling of the Arctic, the simulations of ECBilt-CLIO as well as IPSL are similar to our results, while TraCE-21ka differs from our results. In particular, both ECBilt-CLIO and IPSL simulations show asymmetric cooling in the two Arctic regions and both have a greater cooling in the North Atlantic than in the North Pacific. The first transient simulations (Timm and Timmermann, 2007) based on the ECBilt-CLIO model with a horizontal resolution of about 5.6° × 5.5° and covering the past 21 ka, with the external forcing of ice cover, greenhouse gas concentration, and orbital configuration. The unaccelerated simulations using ECBilt-CLIO similarly reveal a regional asymmetry in temperature variability during the early-mid to late Holocene, with greater cooling in the North Atlantic (-0.26 °C cooling) than in the North Pacific (-0.02 °C cooling) (significant at the 90% confidence level; Supplementary Fig. 3). The second nonaccelerated simulations based on IPSL ESM model (Braconnot et al., 2019) explored the relationship between climate change and vegetation over the past 6000 years. The IPSL ESM also captures the asymmetry characteristics of Arctic temperature variability between the mid Holocene (4-6ka BP) and late Holocene (0-2 ka BP),with -0,14 °C cooling in the Pacific Arctic and -0.18 °C cooling in the Atlantic Arctic (Supplementary Fig. 3). However, the result of IPSL did not significant. The asymmetric changes are more pronounced if we focus only on the North Atlantic and North Pacific ocean regions. However, the results of TraCE-21ka are slightly different. TraCE-21ka is forced by the Earth's orbital parameters forcing, the greenhouse gas forcing, the meltwater flux forcing, and the continental ice sheets forcing. Its results show that the difference in annual

mean temperature between the early-mid Holocene and late Holocene is significantly cooler in the Arctic and there is a regional asymmetry in temperature changes between the two regions, with -0,94 °C cooling in the Pacific Arctic and -0.06 °C cooling in the Atlantic Arctic (significant at the 90% confidence level; Supplementary Fig. 3). However, unlike NNU-Hol, the cooling in the Atlantic Arctic is smaller in magnitude than that in the Pacific Arctic in the asymmetric change of temperature in TraCE-21ka. This result might be attributed to the different external forcings of TraCE-21ka and NNU-Hol: there is still considerable residual ice-sheets and freshwater discharge in TraCE-21ka at about 8 ka BP which is not present in NNU-Hol.

Earlier work was more focused on the effects of the Arctic amplification in recent decades. Based on the observation data, it is assumed that AA will further increase the possibility of extreme weather in mid-latitude regions by increasing the temperature gradient from the equator to the pole in dynamic ways. However, the mechanism behind this is still not fully understood (Xue et al., 2017; Cohen et al., 2018; Chen et al., 2015; Vavrus, 2018; Screen, 2017). For long-term Arctic climate research, it is mainly the study of temperature trend changes based on reconstructed proxy data (Kaufman et al., 2004; Meyer et al., 2015; Briner et al., 2016). This study provides a additional insights on Holocene time scales into the regional characteristics of Arctic temperature changes. With the addition of our model analysis this study improves our understanding of the drivers of variability changes in Arctic temperature on centennial and millennial time scales. The role of sea ice and sea level pressure offers an explanation for the regional differences in Arctic temperature trends.

The study demonstrates that the asymmetry of the orbital forcing-only simulations is greater than that of the full forcing simulation. Stronger response to orbital forcing implies that the other forcings (e.g. GHG) may compensate this asymmetry. The stronger response to orbital forcing implies that the combined effects of other forcings (e.g., solar irradiance, volcanic eruptions, greenhouse gases, and land use/land cover) and internal climate variability may compensate for the temperature asymmetry pattern. In particular, volcanic eruptions have a larger impact on temperature at a short time scales in region, while the greenhouse gas forcing plays a more critical role as a driver of future climate than orbital forcing. Nevertheless, the individual contribution of each forcing has not been clearly investigated in our study. To fully understand the potential compensation effects induced by volcanic eruption, TSI and GHG forcing, the next steps regarding paleo Arctic temperature studies are compiling more high-resolution, seasonal paleoclimate temperature data, employing and analyzing the results of more comprehensive state-of-the-art simulations of different forcings (e.g. TraCE-21ka (Liu et al., 2014), IPSL (Braconnot et al., 2019)) for reducing the uncertainty of the models and the validation of the roles of different forcing.

## 5 Summary

The findings from this paper suggests that the Arctic temperature has an asymmetric cooling trend with more cooling over the Atlantic Arctic (-1.54 °C) than the Pacific Arctic (-0.61 °C) during the Holocene, based on Temperature 12k database (Kaufman et al., 2020). In our modeling results, a similar asymmetric change of temperature in the Arctic can be reproduced by CESM, dominated by orbital forcing. Our model simulations show the strongest cooling in the Arctic Ocean, Greenland Island, and the North Atlantic, while the temperature in northern Eurasia and northern Canada shows a warming trend. There is a seasonal

difference in the asymmetric cooling trend, which is dominated by the DJF temperature variability. The Arctic dipole mode of sea level pressure and sea ice play a major role in asymmetric temperature changes. Using no-accelerated simulations to test the robustness of our results the Arctic temperature asymmetry, we found that the ECBilt-CLIO and IPSL simulations are similar to our results, while TraCE-21ka shows the opposite asymmetry.

**Acknowledgments**

This work was jointly supported by the National Natural Science Foundation of China (Grant Nos. 42130604, 42105044, 41971108, 42105044, and 42111530182) and the Priority Academic Program Development of Jiangsu Higher Education Institutions (Grant No. 164320H116). J. Sjolte was supported by the strategic research program of ModEling the Regional and Global Earth system (MERGE) hosted by the Faculty of Science at Lund University. Z. Lu received funding from FORMAS mobility (Grant no. 2020-02267)

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
