# Peer review of "Asymmetric changes of temperature in the Arctic during the Holocene based on a transient run with the CESM"

_Climate of the Past, 2022_

## Author Comment (AC1)

Review of cp-2022-22: "Asymmetric changes of temperature in the Arctic during the Holocene based on a transient run with the CESM" by Zhang et al.
* * *
Summary

This paper argues that there was an asymmetric temperature change between the Atlantic and Pacific sectors of the Arctic from the mid- to late-Holocene. The authors find this pattern in the temp12k global Holocene temperature reconstruction and also in transient climate model simulations with CESM. They argue that this is caused by orbital modulation of the Artic Dipole pattern and the Pacific Decadal Oscillation.
* * *
Main comments:

The paper presents an interesting hypothesis with a lot of analyses to back up the main results. However, in places it seems like the results need to be better supported with evaluation of the uncertainties, while the link to the modes of variability may benefit from more elaboration. My main comments are as follows:

Reply: Thank you very much for your valuable suggestions and comments on our manuscript. We have carefully considered the comments and tried our best to address every one of them.

1) If I understood correctly, this Holocene simulation (AF) does not include changes in ice-sheet/sea-level forcing? It so, this might be an important caveat for the response in the Arctic. Although the global sea-level has stabilised by around 6 ka BP, this is right in the middle of the early-Holocene time window that you analyse throughout. I think some discussion of this is needed.

Reply: Thank you for your comments and suggestions. Yes, you understand correctly. The simulation doesn't include the changes of boundary conditions of ice-sheet and sea-level. It's indeed worth a discussion in the manuscript. We will add a discussion to point this out.

2) A more robust evaluation of the proxy-based signal is needed in section 3.1. The asymmetry is dependent on a relatively small number of points that show a stronger cooling in the Atlantic sector of Figure 1. If the coolest 2-3 of these were removed it looks like the asymmetry could likewise disappear. This makes one wonder whether the asymmetry is an artefact of the limited coverage by the proxies? Could you evaluate this in more detail? Perhaps add a histogram of the reconstructed temperature changes in the two regions?

Reply: Thank you for your comments and suggestions. The histogram of the reconstructed temperature changes is shown below. It looks more intuitive that way. There are two reconstruction records showing extreme cooling (-10.1℃) and warming (7.8℃) respectively. Although there are individual reconstructions that are outside the normal range, it is not the case that the asymmetric changes in temperature would disappears when 2-3 coldest proxies in the Atlantic Arctic region are removed. Removing individual values that are out of the norm, overall, the temperature asymmetry in the two regions is still robust (p<0.10).

[Figure]

3)The reconstructed and simulated regional temperature anomalies are given to 2 decimal places which feels overly-precise. It would be more convincing if the estimated uncertainties on these values were presented.

Reply: Thank you for your comments and suggestions. We used box figure (see below) as well as t-test to help estimate the significance of temperature anomalies. Due to the small sample size of the Pacific Arctic reconstruction data, the temperature changes are only showing significance (p<0.10) on t-test.

[Figure]

4) Assuming that the reconstructed asymmetry is robust to the choice of points it is not clear on first reading that the model actually replicates the 'asymmetric' temperature response in the annual mean as only the separate seasons are shown. Since the proxies are calibrated to reflect the annual mean signal I think it would be beneficial to show the annual-mean model result.

Reply: Thank you for your comments and suggestions. We will add the annual-mean model result in temperature anomalies. The figure below shows the asymmetric temperature changes in annual-mean model output. Similarly to proxy data, it depicts the difference in temperature variation between the two regions, with cooling of -1.0℃ and -0.64℃ in the Atlantic Arctic and Pacific Arctic, respectively.

[Figure]

5) The analysis of the atmospheric dynamics is not easy to follow (see comments below) and it is difficult to understand precisely how the PDO/AD modes combine to produce the seasonal-mean signal in the sea-ice.

Reply: Thank you for your comments and suggestions. We apologize for not making it clearer and more understandable. We will reorganize this section in the revised manuscript. In short, the main forcing processes are: 1) the PDO's potential phase dominates the SLP, it affects the AD mode; 2) AD mode brings in warm southerly winds along the shores of the East Siberian and Chukchi seas. It favors strong sea-ice melt in these sectors and pushes the ice away from the coast, leaving open water; 3) The pressure pattern also favors the transport of sea-ice out of the Arctic Ocean and into the North Atlantic through Fram Strait. In turn, it contributes to the asymmetric change in temperature in Arctic.

6) Changes in ocean circulation are not mentioned, but given they are important for the past 2000 years (Zhong et al 2018), it would be worth evaluating.

Reply: Thank you for your comments and suggestions. Zhong et al (2018) proposed that changes in ocean salinity leading to an increase in ocean density, which further affects the transport of heat in the northern North Atlantic and contributes to asymmetric temperature changes. Our manuscript mainly focuses on the analysis of the role of atmospheric dynamics and sea ice. In the revised manuscript, we will discuss the effect of ocean circulation on asymmetric temperature changes more in the discussion section in the context of the importance of ocean circulation.
* * *
Minor comments:

Line 102: Is the Glimmer ice sheet model used in this study or is it deactivated?

Reply: Thank you for your comments and suggestions. The Glimmer ice sheet model is coupled in the model simulations. We will clarify that in the text.

Line 103: I think you should cite Hurrell et al 2013, instead of this web link.

Reply: Thank you for your comments and suggestions. We have modified it.

Line 109: It's not clear how the Gao et al reconstruction is used for the Holocene as in their paper they only discuss the last 1000 years. Please could you expand on this?

Reply: Thank you for your comments and suggestions. The reconstruction data for volcanoes during the Holocene have not yet been published, and we have revised this citation.

Line 113: I could not find Wan et al. (2020) in the reference list.

Reply: Thank you for your comments and suggestions. We've added it.

Line 138: This link does not appear to describe the Jonkers et al 2020 dataset or anything else that is mentioned in this manuscript.

Reply: Thank you for your comments and suggestions. The link is miss one number. We've modified it as "http://www.ncdc.noaa.gov/paleo/study/27330".

Line 149: "... with red indicating an increase in temperature between the late and the early-mid Holocene (0-2 ka BP and 5-8 ka BP), while the blue indicating and decreasing." This can be omitted.

Reply: Thank you for your comments and suggestions. We've removed it.

Line 154-155: These values to 2 decimal places seem overly precise. Please could you estimate the uncertainty in these two values?

Reply: Thank you for your comments and suggestions. We've modified it. Due to the small sample size of the Pacific Arctic reconstruction data, the temperature changes are only showing significance (p<0.10) on t-test.

Line 173: again the regional average temperature anomalies should include uncertainties. I suspect 2 decimal places is overly-precise.

Reply: Thank you for your comments and suggestions. We've modified it. The temperature changes in two region showed significance (p<0.01) in t-test.

Line 206: This sentence starting "Many studies" makes it soucnd like these are all studies on the Holocene, but I believe that they are all focussed on the present-day. Please re-word to clarify this.

Reply: Thank you for your comments and suggestions. We've modified it.

Line 223-227: "The difference in SLP between the two periods does show a similar dipole pattern, but combined with the stronger SLP in the late Holocene than in the early-mid Holocene shown above, it can be assumed that the stronger Arctic dipole in the late period had a greater role in influencing sea ice" Perhaps I have missed something, but I don't follow this.

Reply: Thank you for your comments and suggestions. The manuscript we want to express that the Arctic dipole mode in the late Holocene is stronger than that in early-mid Holocene, and the atmospheric circulation under its influence should also have a stronger influence on sea ice. We have revised this description to make it easier to read.

Lines 236-249: It's not clear how the regressed UV winds and sea-ice on PC2 are responsible for the climatological signal. I think this needs to be elaborated on.

Reply: Thank you for your comments and suggestions. We want to illustrate by changes in UV and sea ice that changes in the SLP brings in warm southerly winds along the shores of the East Siberian and Chukchi seas. It favors strong ice melt in these sectors and pushes the ice away from the coast, leaving open water. The pressure pattern also favors the transport of ice out of the Arctic Ocean and into the North Atlantic through Fram Strait. Differences in the distribution of sea ice in turn change the heat balance through feedback, leading to inconsistent changes in Arctic temperatures. We will add more details in the text to make this part more readable.

Line 260: "The index indicates that negative PDO dominates the late Holocene, while the positive and negative PDO phases oscillate during the early-mid Holocene." This is not clear from the figure. Please can you provide a statistic that shows this.

Reply: Thank you for your comments and suggestions. We've added this analysis. The Negative phase PDO dominated the late Holocene, accounting for more than 90%. On the other hand, the PDO in the positive phase accounted for 62% in the mid Holocene.

Lines 265, 267: Please specify what you are comparing with this spatial correlation coefficient?

Reply: Thank you for your comments and suggestions. We compare the EOF leading pattern of SLP in the whole early-mid Holocene and the EOF leading pattern of SLP in the positive PDO year in the early-mid Holocene, with the spatial correlation coefficient is 0.96. And We compare the EOF leading pattern of SLP in the whole late Holocene and the EOF leading pattern of SLP in the negative PDO year in the late Holocene, with the spatial correlation coefficient is 0.99. We've modified this description.

Line 280: Your results mirror findings of Zhong et al 2018. However, they invoked a significant role of the ocean circulation. Is that important in the present model results?

Reply: Thank you for your comments and suggestions. Yes, the ocean circulation is another important aspect in Zhong et al 2018. In this article, our analysis focuses more on the importance of atmospheric dynamics and sea ice. We will add some ocean circulation analyses in the discussion section and compare out results with Zhang et al. 2018.
* * *
Comments on the figures:

Throughout the labels on figures could be tailored for easier reading of the figures. As it is one has to read the caption carefully to understand what the multi-panelled figures are showing.

Reply: Thank you for your comments and suggestions. We will modify it.

Figure 1: For clarity could you include in this caption whether this is late Holocene minus early Holocene?

Reply: Thank you for your comments and suggestions. We've modified it.

Figure 3: I would like to see the annual-mean model result as the proxies are calibrated to this if I understand correctly?

Reply: Thank you for your comments and suggestions. The annual-mean shows similar results. We've added it.

Figure 6: It would probably be helpful to have the same y-axis limits on panels (c) and (d). Also, are the timeseries of the PC 2 smoothed?

Reply: Thank you for your comments and suggestions. Yes, they are smoothed. But when regressing based on the PC time series we use the original unfiltered PC time series. This needs to be made clear. We've modified it.

Figure 9: is this the AF or the ORBIT-only simulation? Do they both look similar?

Reply: Thank you for your comments and suggestions. It is the AF simulation. Yes, they look similar.
* * *
Technical corrections:

Line 148: "while the blue indicating and decreasing."   Typo here.

Reply: Thank you for your comments and suggestions. We've modified it.

Figure 10: The captions says EOF1 but the figure labels say EOF2. I assume they should both same EOF1?

Reply: Thank you for your comments and suggestions. We've modified it.
* * *
References:

Hurrell, J et al (2013). The Community Earth System Model: A Framework for Collaborative Research, Bull Am Met Soc, 94,9, https://doi.org/10.1175/BAMS-D-12-00121.1.

Wan Lingfeng, Liu Jian, Gao Chaochao, Sun Weiyi, Ning Liang, Yan Mi. Study about influence of the Holocene volcanic eruptions on temperature variation trend by simulation[J]. Quaternary Sciences, 2020, 40(6): 1597-1610. doi: 10.11928/j.issn.1001-7410.2020.06.19

---

## Author Comment (AC2)

Review for the manuscript

Asymmetric changes of temperature in the Arctic during the Holocene based on a transient run with the CESM

by Hongyue Zhang et al.

Submitted for publication in Climate of the Past

**General**

The manuscript investigates Arctic temperature changes in an accelerated earth system model (ESM) simulation for the Holocene with CESM. The authors present asymmetric temperature changes between the Pacific and Atlantic parts of the Arctic and attribute those changes to varying pattern of atmospheric circulation and sea ice concentrations. Moreover, authors suggest that those asymmetric changes are especially pronounced in a simulation that is only driven with changes in orbital forcings.

The manuscript is unfortunately not representing the state-of-the-art literature and more important, lacks of simulations that are currently available for the Holocene in a transient sense. Accelerated simulations for the Holocene were expedient because of a lack of computing capacities some 20 years ago. Therefore according conclusions, especially on long term changes such as ocean-related sea ice processes can be afflicted with high uncertainties, also in the context of the interpretation with proxy data.

As such I cannot suggest publication of the manuscript in the present form. Below I list a number of suggestions and more recent studies including non-accelerated simulations that can be used for a substantially revised version of the manuscript.

**Specific**

In the following I will just point to the main concerns and how authors might extent and update their investigations taking into account more recent studies and adapting their hypothesis to more ESM/GCM-relevant questions.

Reply: We appreciate you for your precious time in reviewing our paper and providing valuable and insightful comments. We have carefully considered the comments and tried our best to address every one of them.

Introduction:

The introduction lacks at least one paragraph motivating recent modeling studies over the Holocene, the challenges and implications e.g. of accelerated simulations vs. non-accelerated and the uncertainties involved in reconstructing external drivers (specifically solar and volcanic) for decadal-to-multi- decadal variability (cf. also studies listed as additional references below)

Reply: Thank you for pointing out the potential caveats of the acceleration and uncertainties in our results that were not fully discussed in the previous manuscript. Indeed, it is important to have more discussions on the recent modeling studies and comparisons between accelerated and non-accelerated simulations. For instance, Varma et al. (2012) compared the simulation results with 10 times acceleration and non-acceleration, and found that there is no significant difference in the characteristics of global surface climate change. Timm and Timmermann (2007) used the ECBilt-CLIO model to simulate the climate since the Last Glacial Maximum (LGM) by 10 times acceleration, and compared the simulation results without acceleration and found that the simulation results with 10 times acceleration reproduced well the large-scale trend of atmospheric temperature in the Holocene. Lu et al. (2019) found that the acceleration leads to suppressed and delayed responses mainly in the deep sea and has less robust effect on the surface and subsurface. Jing et al. (2022) compared the temperature and precipitation changes in NNU-Holocene simulation and Trace-21k non-acceleration simulation, and in terms of overall trend and distribution, the temperature and precipitation distribution patterns of NNU and Trace are similar. These and the uncertainties of reconstructing external drivers will give the reader a more complete understanding of the motivation of our study. It should be pointed out that we have focused more on the long-term climate change (linear trend) rather than decadal to multi-decadal timescale changes. Thank you for the additional references. We will add these descriptions in the introduction section.

[Figure]

Fig. 12 Time sequence of the vertical temperature profile in a simple diffusion model under three acceleration scenarios: (upper panel) 100-fold acceleration, (middle panel) tenfold acceleration and (bottom panel) non-acceleration. (from Lu et al. 2019)

Another crucial and yet missing part is on the potential drivers giving rise to an asymmetric temperature response. Some mechanisms such as changes in equator-to-pole temperature gradient and/or changes in overall sea ice concentrations are presented. But no hypothesis or guiding question in how those general changes should result in regionally different responses are discussed.

Reply: Thank you for pointing this out. We will add some detailed discussion of the mechanisms on how these potential drivers might lead to the regionally different responses.

2 Method and data

2.1 The CESM model and the transient simulations

ll. 106 ff: The authors describe their acceleration technique, also using changes in solar and volcanic output. I was wondering how those changes, reconstructed on yearly time scales can be implemented in a simulation with an acceleration factor of 10. (e.g. typically more than 2 volcanic eruptions happen per decade). How is this temporal discrepancy between annual reconstructions for accelerated simulations accounted for, also considering the post-volcanic effects on the simulated climate.

Reply: Thank you for your important comments. We aggregate the solar forcing to annual timescale, and then do a 10-year average as the time series of solar forcing used in the simulation is shown in Figure below (Wan et al. 2020). For the volcanic forcing, the volcanic events during the 10-year period were integrated into one volcanic eruption event. On the basis of this assumption, the horizontal diffusion of lower stratospheric aerosols was calculated using the stratospheric transport parameters. Based on the stratospheric-tropospheric folding and BD (Brewer Dobson) circulation theory latitude- and time-dependent functions to describe aerosol production and deposition (Grieser J et al.1999; Holton J et al. 1995). The details of the modelling methods will be added to the revised manuscript. Because we focus on long-time-scale changes, and volcanic eruptions are found to have a smaller impact on climate than orbital forcing. Therefore, we mainly investigate the orbital forcing effects in All forcing simulation and ORB simulation.

[Figure]

Fig. 1 The external forcing timeseries used in the NNU-Hol simulation. The TSI VOL, GHG and LUCC are a b c and d respectively (Wan et al.2020)

ll. 116 ff: There are new, and non-accelerated comprehensive Earth System model simulations available (cf. references) that should be used as additional source of information to back-up results based on the accelerated simulations with CESM.

Reply: Thank you for your comments and suggestions. We found that similar temperature asymmetry changes also exists in the Arctic region based on the results of Trace-21k non-acceleration simulation. We will consider to add the Trace results to the manuscript.

Another general comment relates to the questions why the authors did not at least use an ensemble approach for their simulations to estimate the amount of long-term (centennial-to-millennial scale) climate variability.

Reply: Thank you for your comments and suggestions. Unfortunately, the main restriction is because of limited computing resources. For our long-term (12ka) climate simulations, with multiple forcings applied, we employ the acceleration technique, and each simulation has only one member.

2.2 Reconstructing Paleo Proxy data

This paragraph just lists the proxy data sets used for comparison without any information on potential uncertainties involved in the reconstructions, e.g. related to the uncertainties in the proxy archives towards their meteorological/climate variables, dating uncertainties, regional sparseness of proxy data, especially in the Arctic domain.

Reply: Thank you for your comments and suggestions. We will revise the manuscript to add a part of the uncertainty description. It should be noted that the uncertainty of the reconstruction has already been discussed in Kaufman et al. (2020). Since this is not the main focus of our study, we did not discuss the uncertainties in the proxy results in details.

Since the authors investigate changes in ocean-related sea ice variability, also a paragraph on proxies representing changes in sea-ice concentrations including their uncertainties would be helpful.

Reply: Thank you for your comments and suggestions. We haven't included the proxies of sea ice yet. It's a good suggestion, we will look for some relevant sea ice reconstructions and compare with our model results.

3. Result

3.1 Arctic temperature change

ll. 152 ff: How robust are the temperature changes? Are they statistically significantly different to internal changes. Therefore, applying a statistical test is helpful to estimate the amount of internal variability between the two different periods, preferentially taking into account the serial correlations within the proxy-based estimations of temperature variability.

Reply: Thank you for your comments and suggestions. We used box figure (see below) as well as t-tests to help estimate the amount of internal variability between the two different periods. Due to the small sample size of the Pacific Arctic reconstruction data, the temperature changes are only showing significance (p<0.10) on t-test.

[Figure]

ll. 172 ff: How significant are the changes between the Arctic and the Pacific region ? (i.e. -0.67 vs. +0.09.) Especially the Pacific trend seems to be statistically indistinguishable from a zero trend).

Reply: Thank you for your comments and suggestions. The t-test suggests the temperature changes in two region are significant ($p<0.01$). For winter temperature change it seems to be statistically indistinguishable from a zero trend, but for annual average or summer, there is a significant cooling.

ll. 191 ff: Also for the model-based differences of the sea ice a local statistical test on the spatial pattern including the effect of serial correlation is important to test the robustness and statistical significance of the according changes.

Reply: Thank you for your comments and suggestions. We will modify the figure and perform spatial significance test for the figure of sea ice change.

ll. 202 ff: Changes in atmospheric circulation are also influenced to a high degree to internal variability – as such it is very important to use additional model simulations to back-up those changes, resulting from the CESM accelerated simulation. Moreover, why are the results of the orbital simulation are "more significant" than the one for the all forcings ? On Holocene time scales changes in orbital forcing on seasonal time scales exert a larger impact than the decadal-and sub-decadal changes caused by solar and volcanic activity. Therefore it is important to describe in greater detail how changes in solar and volcanic forcings are implemented into the accelerated CESM simulation.

Reply: Thank you for your comments and suggestions. As mentioned above, the difference between acceleration and non-acceleration simulation is the dampened and delayed response to external forcings in the deep ocean for the latter. There should be no big differences for the atmospheric circulation and surface climate response. We will consider to add some results of other non-accelerated simulations (e.g. Trace-21k). Orbital forcing as the most obvious driver of long-term trend changes during the Holocene. Volcanic and TSI forcing have less impacts on long-term trends, and their role is more dominant on shorter timescales such as decadal and multi- decadal scale. However, the aim of our study is not to focus on these shorter timescales, so our analyses focus on orbital forcing and All forcing simulations.

3.3 EOF of SLP and UV wind regression and 3.4 The connection between Arctic dipole pattern and PDO

The whole sections lack a more thorough motivation on i) how the statistical concepts are used/defined and the ii) the robustness and statistical significance of the according regression patterns between the PCs and the underlying wind/sea ice fields. For instance, the PCs presented in Fig. 6 are (obviously) filtered with a low-pass filter. This should be accounted for when discussing and presenting the results.

Reply: Thank you for your comments and suggestions. We will add more statistical tests. As the PCs presented in Fig. 6, you're right, it's filtered. We will clarify the filter we used in the revised manuscript.

Further, in addition to the UV regression, a Canonical correlation analysis would be better suited for this kind of investigation in section 3.3, since the rationale is to compare the common behavior of patterns (in this case the spatially resolved SLP and wind/sea ice fields.)

Reply: Thank you for your comments and suggestions. We will try the Canonical correlation analysis methods to compare with that.

A last point is again on the validity and model-dependence of the results based only on the accelerated simulation with CESM. This is in my opinion the weakest but most crucial point of the study.

Reply: Thank you for your comments and suggestions. As mentioned before, we will add some describing the validation of CESM simulations and analyze some results of unaccelerated simulations, but the main focus will remain on the CESM results.

4. Discussion

l. 291: Authors should formulate more nuanced that in this very version of the manuscript, results only apply to their few accelerated simulations with CESM that need to be compared with more recent, non-accelerated studies.

Reply: Thank you for your comments and suggestions. We will add a comparison with more recent, non-accelerated studies.

l. 293: How should GHG changes, only changing very moderately in the pre-industrial period of the Holocene counteract any changes in orbital forcing ? If any, volcanic (and maybe in parts) negative periods of solar activity could counteract the negative trend in orbital forcing during the JJA season over the Arctic.

Reply: Thank you for your comments and suggestions. We will revise this statement. We mean that the orbital simulation shows stronger asymmetric changes compare to the All forcing simulation. This implies that the combined effect of other forcings (solar irradiance, volcanic eruptions, greenhouse gases, and land use/land cover) and internal climate variability is offsetting this asymmetric changes (as opposite to the orital forcing). The contribution of different forcings is what we will need to study in the future. The reason why we mention "e.g. GHG" is that in future climate change, the GHG is an important factor that cannot be ignored. We will revise this paragraph to make it more clearly.

l. 284: The authors state that additional simulations should be used for investigations. Since those simulations are yet available authors should use them as an integral part of their revised study and thoroughly test their hypotheses with non-accelerated simulations and those carried out with different CMIP4-types of models.

Reply: Thank you for your comments and suggestions. We will add a comparison with more recent, non-accelerated studies.

Figures:

Fig 3.1: How does the Proxy (z-score) and the Model (°C) compare on the same axis ? In my opinion it would be necessary to show both on the same scale for an appropriate comparison.

Reply: Thank you for your comments and suggestions. We've modified it.

Fig. 5: Please use units of hPa when presenting changes of sea level pressure fields.

Reply: Thank you for your comments and suggestions. We've modified it.

Fig 6, 9a and 10a: In this form of the presentation, the EOF pattern seem to carry normalized values (i.e. z-scores). In order to re-normalize the EOFs (i.e. eigenvectors), the patterns should be multiplied with the square root of their eigenvalue. Then the EOF patterns carry the units (in this case Pa(hPa) for SLP and K for SSTs, respectively). Eventually the according (original) PCs should be divided by the square root of the eigenvalue in order to show consistent patterns between EOFs and PCs. In addition, the temporal filtering should be indicated for the time series.

Reply: Thank you for your comments and suggestions. We will modified it.

Additional references / State-of-the art Holocene ESM simulations:

Transient Holocene simulation (6ka BP - 2ka BP) with interactive vegetation and phenology: https://vesg.ipsl.upmc.fr/thredds/catalog/work/p86mart/IPSLCM6/PROD/ Holocene/TR6AV-Sr02/catalog.html

Braconnot, P., Zhu, D., Marti, O. and Servonnat, J.: Strengths and challenges for transient Mid- to Late Holocene simulations with dynamical vegetation, Clim. Past, 15(3), 997–1024, doi:10.5194/cp-15-997-2019, 2019

Braconnot, P., Marti, O., Crétat, J., Zhu, D., Sanogo, S., Balkanski, Y., Caubel, A., Cozic, A., Foujols, M.-A. and Servonnat, J.: Transient simulations of the last 6000 years with the IPSL model, in PMIP Workshop group P2FVAR., 2019.

Bader, J., Jungclaus, J., Krivova, N., Lorenz, S., Maycock, A., Raddatz, T., Schmidt, H., Toohey, M., Wu, C.-J. & Claussen, M., 2020: Global temperature modes shed light

on the Holocene temperature conundrum. Nature Communications, 11: 4726. doi:10.1038/s41467-020-18478-6.

Dallmeyer, A., Claussen, M., Lorenz, S. J., Sigl, M., Toohey, M., and Herzschuh, U.: Holocene vegetation transitions and their climatic drivers in MPI-ESM1.2, Clim. Past Discuss. Clim. Past, 17, 2481–2513, https://doi.org/10.5194/cp-17-2481-2021, 2021.

New References:

Varma V, Prange M, Merkel U, et al. Holocene evolution of the Southern Hemisphere westerly winds in transient simulations with global climate models[J]. Climate of the Past, 2012, 8(2): 391-402.

Timm O, Timmerman A (2007) Simulation fo the last 21000 years using accelerated transient boundary conditions*. J Clim 20(17):4377–4401

Lu, Z., Liu, Z. Orbital modulation of ENSO seasonal phase locking. Clim Dyn 52, 4329–4350 (2019). https://doi.org/10.1007/s00382-018-4382-1

Jing Y, Liu J, Wan L. Comparison of climate responses to orbital forcing at different latitudes during the Holocene[J]. Quaternary International, 2022, 622: 65-76.

Holton J R, Haynes P H, McIntyre M E, et al. Stratosphere-troposphere exchange[J]. Reviews of geophysics, 1995, 33(4): 403-439.

GRIESER J, SCHONWIESE C D. Parameterization of spatio-temporal patterns of volcanic aerosol induced stratospheric optical depth and its climate radiative forcing[J]. Atmósfera, 1999, 12(2).

---

## Author Response (AR1)

**First Review for the manuscript**

Asymmetric changes of temperature in the Arctic during the Holocene based on a transient run with the CESM

by Hongyue Zhang et al.

Submitted for publication in Climate of the Past
* * *
Summary

This paper argues that there was an asymmetric temperature change between the Atlantic and Pacific sectors of the Arctic from the mid- to late-Holocene. The authors find this pattern in the temp12k global Holocene temperature reconstruction and also in transient climate model simulations with CESM. They argue that this is caused by orbital modulation of the Artic Dipole pattern and the Pacific Decadal Oscillation.
* * *
Main comments:

The paper presents an interesting hypothesis with a lot of analyses to back up the main results. However, in places it seems like the results need to be better supported with evaluation of the uncertainties, while the link to the modes of variability may benefit from more elaboration. My main comments are as follows:

Reply: Thank you very much for your valuable suggestions and comments on our manuscript. We have carefully considered the comments and tried our best to address every one of them.

1) If I understood correctly, this Holocene simulation (AF) does not include changes in ice-sheet/sea-level forcing? It so, this might be an important caveat for the response in the Arctic. Although the global sea-level has stabilised by around 6 ka BP, this is right in the middle of the early-Holocene time window that you analyse throughout. I think some discussion of this is needed.

Reply: Thank you for your comments and suggestions. Yes, you understand correctly. The simulation doesn't include the changes of boundary conditions of ice-sheet and sea-level. We highlighted this in the model data introduction. It's indeed worth a discussion in the manuscript. In fact, we believe that ice-sheet and meltwater flux are the reason for the discrepancy between TraCE-21ka and NNU-Hol, and we describe this in the Discussion section.

2) A more robust evaluation of the proxy-based signal is needed in section 3.1. The asymmetry is dependent on a relatively small number of points that show a stronger cooling in the Atlantic sector of Figure 1. If the coolest 2-3 of these were removed it looks like the asymmetry could likewise disappear. This makes one wonder whether the asymmetry is an artefact of the limited coverage by the proxies? Could you evaluate this in more detail? Perhaps add a histogram of the reconstructed temperature changes in the two regions?

Reply: As suggested by the reviewer, we added histograms of temperature changes for each proxy site and calculated significance to indicate the robustness of the temperature anomaly feature. The histogram of the reconstructed temperature changes is shown below. It looks more intuitive that way. There are two reconstruction records showing extreme cooling (-10.1 ℃) and warming (7.8 ℃) respectively. Although there are individual reconstructions that are outside the normal range, it is not the case that the asymmetric changes in temperature would disappears when 2-3 coldest proxies in the Atlantic Arctic region are removed. Removing two individual values that are out of the norm, overall, the temperature asymmetry in the two regions is still robust (p<0.10).

[Figure]

3)The reconstructed and simulated regional temperature anomalies are given to 2 decimal places which feels overly-precise. It would be more convincing if the estimated uncertainties on these values were presented.

Reply: Thank you for your rigorous comment. We used box figure (see below) as well as t-test to help estimate the significance of temperature anomalies. Due to the small sample size of the Pacific Arctic reconstruction data, the temperature changes are only showing significance (p<0.10) on t-test.

[Figure]

4) Assuming that the reconstructed asymmetry is robust to the choice of points it is not clear on first reading that the model actually replicates the 'asymmetric' temperature response in the annual mean as only the separate seasons are shown. Since the proxies are calibrated to reflect the annual mean signal I think it would be beneficial to show the annual-mean model result.

Reply: Thank you for your suggestion, which will make the results more convincing. We add the temperature anomaly annual average based on NNU-Hol in the supplemental material (Fig S3). The figure below shows the asymmetric temperature changes in annual-mean model output. Similarly to proxy data, it depicts the difference in temperature variation between the two regions, with cooling of -1.0℃ and -0.64℃ in the Atlantic Arctic and Pacific Arctic, respectively.

[Figure]

5) The analysis of the atmospheric dynamics is not easy to follow (see comments below) and it is difficult to understand precisely how the PDO/AD modes combine to produce the seasonal-mean signal in the sea-ice.

Reply: Thank you for your comments. We apologize for not making it clearer and more understandable. We reorganized this section in the revised manuscript. In short, the main forcing processes are: 1) the PDO's potential phase dominates the SLP, it affects the AD mode; 2) AD mode brings in warm southerly winds along the shores of the East Siberian and Chukchi seas. It favors strong sea-ice melt in these sectors and pushes the ice away from the coast, leaving open water; 3) The pressure pattern also favors the transport of sea-ice out of the Arctic Ocean and into the North Atlantic through Fram Strait. In turn, it contributes to the asymmetric change in temperature in Arctic.

6) Changes in ocean circulation are not mentioned, but given they are important for the past 2000 years (Zhong et al 2018), it would be worth evaluating.

Reply: Thank you for your comments and suggestions. Zhong et al (2018) proposed that changes in ocean salinity leading to an increase in ocean density, which further affects the transport of heat in the northern North Atlantic and contributes to asymmetric temperature changes. Our manuscript mainly focuses on the analysis of the role of atmospheric dynamics and sea ice. In the revised manuscript, we discussed the effect of ocean circulation on asymmetric temperature changes more in the discussion section in the context of the importance of ocean circulation. "Zhong et al.(2018) believes that the change in ocean density due to the millennial cooling has led to a slowdown of the subpolar circulation. This deceleration of the subpolar circulation reduces the heat advection in the northern North Atlantic and intensifies the cooling in Alatntic Arctic region. Our results are in line with the hypothesis that sea level pressure and sea ice play the important role in asymmetric cooling. However, there is no freshwater forcing in the NNU-Hol simulation, and further studies are needed to clarify how the mechanism we have identified in this work is related to the work of Zhong et al. (2018)."
* * *
Minor comments:

Line 102: Is the Glimmer ice sheet model used in this study or is it deactivated?

Reply: Thank you for pointing out this problem in manuscript. The Glimmer ice sheet model is deactivated in the model simulations. We clarify that in the text .

Line 103: I think you should cite Hurrell et al 2013, instead of this web link.

Reply: Thank you for your suggestions. We have modified it.

Line 109: It's not clear how the Gao et al reconstruction is used for the Holocene as in their paper they only discuss the last 1000 years. Please could you expand on this?

Reply: Thank you for your comments. The reconstruction data for volcanoes during the Holocene have not yet been published, and we have revised this citation.

Line 113: I could not find Wan et al. (2020) in the reference list.

Reply: Thank you so much for your careful check. We've added it.

Line 138: This link does not appear to describe the Jonkers et al 2020 dataset or anything else that is mentioned in this manuscript.

Reply: Thank you so much for your careful check. The link is miss one number. We've modified it as "http://www.ncdc.noaa.gov/paleo/study/27330".

Line 149: "... with red indicating an increase in temperature between the late and the early-mid Holocene (0-2 ka BP and 5-8 ka BP), while the blue indicating and decreasing." This can be omitted.

Reply: Thank you for your suggestions. We've removed it.

Line 154-155: These values to 2 decimal places seem overly precise. Please could you estimate the uncertainty in these two values?

Reply: Thank you for pointing out this in manuscript. We've modified it. Due to the small sample size of the Pacific Arctic reconstruction data, the temperature changes are only showing significance ($p<0.10$) on t-test. We also represent its robustness through box plots.

Line 173: again the regional average temperature anomalies should include uncertainties. I suspect 2 decimal places is overly-precise.

Reply: Thank you for your suggestions. We've modified it. The temperature changes in two region showed significance ($p<0.01$) in t-test.

Line 206: This sentence starting "Many studies" makes it soucnd like these are all studies on the Holocene, but I believe that they are all focussed on the present-day. Please re-word to clarify this.

Reply: Thank you for pointing out this in manuscript. We re-emphasize that these are study about the present-day.

Line 223-227: "The difference in SLP between the two periods does show a similar dipole pattern, but combined with the stronger SLP in the late Holocene than in the early-mid Holocene shown above, it can be assumed that the stronger Arctic dipole in the late period had a greater role in influencing sea ice" Perhaps I have missed something, but I don't follow this.

Reply: We feel sorry for the reading inconvenience brought to the reviewer. The manuscript we want to express that the Arctic dipole mode in the late Holocene is stronger than that in early-mid Holocene, and the atmospheric circulation under its influence should also have a stronger influence on sea ice. We have revised this description to "Combined with the stronger SLP in the late Holocene shown above Figure 5, it can be assumed that the late Holocene has a stronger Arctic dipole pattern. By regressing the sea ice distribution onto the second Principal Component time series (Fig.7), we show that the Arctic dipole in the late Holocene had a greater role in influencing sea ice. Therefore, the intensity of the dipole mode appears to dominate the sea-ice distribution in the Arctic, and the hypothesis can be verified."

Lines 236-249: It's not clear how the regressed UV winds and sea-ice on PC2 are responsible for the climatological signal. I think this needs to be elaborated on.

Reply: Thank you for your comments. We want to illustrate by changes in UV and sea ice that changes in the SLP brings in warm southerly winds along the shores of the East Siberian and Chukchi seas. Figure 7 shows the percentage of change in sea ice related to the variability of PC2 of SLP. It favors strong ice melt in these sectors and pushes the ice away from the coast, leaving open water. The pressure pattern also favors the transport of ice out of the Arctic Ocean and into the North Atlantic through Fram Strait. Warm southerly winds hinder the Holocene cooling trend in the Pacific Arctic. Differences in the distribution of sea ice in turn change the heat balance through feedback, leading to inconsistent changes in Arctic temperatures. We have rewritten this paragraph to make this part more readable.

Line 260: "The index indicates that negative PDO dominates the late Holocene, while the positive and negative PDO phases oscillate during the early-mid Holocene." This is not clear from the figure. Please can you provide a statistic that shows this.

Reply: Thank you for your suggestions. We've re-emphasized this point in the text. The Negative phase PDO dominated the late Holocene, accounting for more than 90%. On the other hand, the PDO in the positive phase accounted for 62% in the mid Holocene.

Lines 265, 267: Please specify what you are comparing with this spatial correlation coefficient?

Reply: Thank you for your suggestions. We compare the EOF leading pattern of SLP in the whole early-mid Holocene and the EOF leading pattern of SLP in the positive PDO year in the early-mid Holocene, with the spatial correlation coefficient is 0.96. And We compare the EOF leading pattern of SLP in the whole late Holocene and the EOF leading pattern of SLP in the negative PDO year in the late Holocene, with the spatial correlation coefficient is 0.99. We've modified this description.

Line 280: Your results mirror findings of Zhong et al 2018. However, they invoked a significant role of the ocean circulation. Is that important in the present model results?

Reply: Thank you for your nice advice. Yes, the ocean circulation is another important aspect in Zhong et al 2018. We have modified the Discussion section to clarify this point. See also reply for Main Comment 6.
* * *
Comments on the figures:

Throughout the labels on figures could be tailored for easier reading of the figures. As it is one has to read the caption carefully to understand what the multi-panelled figures are showing.

Reply: Thank you for your suggestion. We've modified it.

Figure 1: For clarity could you include in this caption whether this is late Holocene minus early Holocene?

Reply: Thank you for your nice suggestion. We've modified it.

Figure 3: I would like to see the annual-mean model result as the proxies are calibrated to this if I understand correctly?

Reply: Thank you for your comments. The annual-mean shows similar results. We have added the distribution of annual average temperature anomalies to Fig S3.

Figure 6: It would probably be helpful to have the same y-axis limits on panels (c) and (d). Also, are the timeseries of the PC 2 smoothed?

Reply: T Thank you for pointing out this. Yes, they are smoothed. But when regressing based on the PC time series we use the original unfiltered PC time series. This needs to be made clear. We've modified it.

Figure 9: is this the AF or the ORBIT-only simulation? Do they both look similar?

Reply: Thank you for your comment. It is the AF simulation. Yes, they look similar.
* * *
Technical corrections:

Line 148: "while the blue indicating and decreasing." Typo here.

Reply: Thank you for your suggestion. We've modified it.

Figure 10: The captions says EOF1 but the figure labels say EOF2. I assume they should both same EOF1?

Reply: Thank you for your comments and suggestions. We've modified it.
* * *
References:

Hurrell, J et al (2013). The Community Earth System Model: A Framework for Collaborative Research, Bull Am Met Soc, 94,9, https://doi.org/10.1175/BAMS-D-12-00121.1.

Wan Lingfeng, Liu Jian, Gao Chaochao, Sun Weiyi, Ning Liang, Yan Mi. Study about influence of the Holocene volcanic eruptions on temperature variation trend by simulation[J]. Quaternary Sciences, 2020, 40(6): 1597-1610. doi: 10.11928/j.issn.1001-7410.2020.06.19

===============================================================

Asymmetric changes of temperature in the Arctic during the Holocene based on a transient run with the CESM

by Hongyue Zhang et al.

Submitted for publication in Climate of the Past

**General**

The manuscript investigates Arctic temperature changes in an accelerated earth system model (ESM) simulation for the Holocene with CESM. The authors present asymmetric temperature changes between the Pacific and Atlantic parts of the Arctic and attribute those changes to varying pattern of atmospheric circulation and sea ice concentrations. Moreover, authors suggest that those asymmetric changes are especially pronounced in a simulation that is only driven with changes in orbital forcings.

The manuscript is unfortunately not representing the state-of-the-art literature and more important, lacks of simulations that are currently available for the Holocene in a transient sense. Accelerated simulations for the Holocene were expedient because of a lack of computing capacities some 20 years ago. Therefore according conclusions, especially on long term changes such as ocean-related sea ice processes can be afflicted with high uncertainties, also in the context of the interpretation with proxy data.

As such I cannot suggest publication of the manuscript in the present form. Below I list a number of suggestions and more recent studies including non-accelerated simulations that can be used for a substantially revised version of the manuscript.

**Specific**

In the following I will just point to the main concerns and how authors might extent and update their investigations taking into account more recent studies and adapting their hypothesis to more ESM/GCM-relevant questions.

Reply: We appreciate you for your precious time in reviewing our paper and providing valuable and insightful comments. We have carefully considered the comments and tried our best to address every one of them.

Introduction:

The introduction lacks at least one paragraph motivating recent modeling studies over the Holocene, the challenges and implications e.g. of accelerated simulations vs. non-accelerated and the uncertainties involved in reconstructing external drivers (specifically solar and volcanic) for decadal-to-multi- decadal variability (cf. also studies listed as additional references below)

Reply: Thank you for pointing out the potential caveats of the lack of comparison with the latest model results, as well as the acceleration and uncertainties in our results that were not fully discussed in the previous manuscript. Thanks for providing additional references, we have added some content about the latest model studies in the introduction section and a part of the analysis in the discussion section. Indeed, it is important to have more discussions on the recent modeling studies and comparisons between accelerated and non-accelerated simulations. For instance, Varma et al. (2012) compared the simulation results with 10 times acceleration and non-acceleration, and found that there is no significant difference in the characteristics of global surface climate change. Timm and Timmermann (2007) used the ECBilt-CLIO model to simulate the climate since the Last Glacial Maximum (LGM) by 10 times acceleration, and compared the simulation results without acceleration and found that the simulation results with 10 times acceleration reproduced well the large-scale trend of atmospheric temperature in the Holocene. Lu et al. (2019) found that the acceleration leads to suppressed and delayed responses mainly in the deep sea and has less robust effect on the surface and subsurface. Jing et al. (2022) compared the temperature and precipitation changes in NNU-Holocene simulation and Trace-21k non-acceleration simulation, and in terms of overall trend and distribution, the temperature and precipitation distribution patterns of NNU and Trace are similar. These and the uncertainties of reconstructing external drivers that we add in the data section will give the reader a more complete understanding of the motivation of our study. It should be pointed out that we have focused more on the long-term climate change (linear trend) rather than decadal to multi-decadal timescale changes.

[Figure]

Fig. 12 Time sequence of the vertical temperature profile in a simple diffusion model under three acceleration scenarios: (upper panel) 100-fold acceleration, (middle panel) tenfold acceleration and (bottom panel) non-acceleration. (from Lu et al. 2019)

Another crucial and yet missing part is on the potential drivers giving rise to an asymmetric temperature response. Some mechanisms such as changes in equator-to-pole temperature gradient and/or changes in overall sea ice concentrations are presented. But no hypothesis or guiding question in how those general changes should result in regionally different responses are discussed.

Reply: Thank you for pointing this out. On the Pacific side, a stronger temperature gradient in equator-to-pole temperature increasing the Aleutian Low strengthens, thereby contributing to winter warming around the Bering Sea as the storm system transports warm air to the poles. On the Atlantic region, the sea ice expansion leads to a slowing of the subpolar circulation combined with the sea ice feedbacks, resulting in further cooling in the Atlantic Arctic. We have added the description of the guiding question at the end of the introduction section

2 Method and data

2.1 The CESM model and the transient simulations

ll. 106 ff: The authors describe their acceleration technique, also using changes in solar and volcanic output. I was wondering how those changes, reconstructed on yearly time scales can be implemented in a simulation with an acceleration factor of 10. (e.g. typically more than 2 volcanic eruptions happen per decade). How is this temporal discrepancy between annual reconstructions for accelerated simulations accounted for, also considering the post-volcanic effects on the simulated climate.

Reply: Thank you for your valuable and insightful comments. We aggregate the solar forcing to annual timescale, and then do a 10-year average as the time series of solar forcing used in the simulation is shown in Figure below (Wan et al. 2020). For the volcanic forcing, the volcanic events during the 10-year period were integrated into one volcanic eruption event. On the basis of this assumption, the horizontal diffusion of lower stratospheric aerosols was calculated using the stratospheric transport parameters. Based on the stratospheric-tropospheric folding and BD (Brewer Dobson) circulation theory latitude- and time-dependent functions to describe aerosol production and deposition (Grieser J et al.1999; Holton J et al. 1995). The details of the modelling methods have been added to the revised manuscript. Because we focus on long-time-scale changes, and volcanic eruptions are found to have a smaller impact on climate than orbital forcing. Therefore, we mainly investigate the orbital forcing effects in All forcing simulation and ORB simulation.

[Figure]

Fig. 1 The external forcing timeseries used in the NNU-Hol simulation. The TSI VOL, GHG and LUCC are a b c and d respectively (Wan et al.2020)

ll. 116 ff: There are new, and non-accelerated comprehensive Earth System model simulations available (cf. references) that should be used as additional source of information to back-up results based on the accelerated simulations with CESM.

Reply: Thank you for your very useful suggestions and additional references. We compared three other non-accelerated simulations covering the Holocene period, Trace-21k and ECBilt-CLIO and IPSL, respectively. In the discussion section, we compare the annual mean temperature variation during the Late Holocene and Early-mid Holocene. The results show that both ECBilt-CLIO and IPSL exhibit consistency with the NNU-HOL results, while Trace-21 also exhibits temperature asymmetry but differs from NNU-HOL, which we attribute to the different external forcing added to the simulation.

Another general comment relates to the questions why the authors did not at least use an ensemble approach for their simulations to estimate the amount of long-term (centennial-to-millennial scale) climate variability.

Reply: Thank you for your comment. Unfortunately, the main restriction is because of limited computing resources. For our long-term (12ka) climate simulations, with multiple forcings applied, we employ the acceleration technique, and each simulation has only one member.

2.2 Reconstructing Paleo Proxy data

This paragraph just lists the proxy data sets used for comparison without any information on potential uncertainties involved in the reconstructions, e.g. related to the uncertainties in the proxy archives towards their meteorological/climate variables, dating uncertainties, regional sparseness of proxy data, especially in the Arctic domain.

Reply: Thank you for your important comments and suggestions. We modified the data section to add a description of the uncertainty in reconstruction. It should be noted that the uncertainty of the reconstruction has already been discussed in Kaufman et al. (2020). Since this is not the main focus of our study, we did not discuss the uncertainties in the proxy results in details.

Since the authors investigate changes in ocean-related sea ice variability, also a paragraph on proxies representing changes in sea-ice concentrations including their uncertainties would be helpful.

Reply: Thank you for your good suggestions. We haven't included the proxies of sea ice yet. However, the changes in sea ice concentration in our model results are consistent with the changes of sea ice during the Holocene in previous research papers. Overall, Arctic sea ice concentrations were low during the early to middle Holocene and increased during the late Holocene cum Neoglacial. Meanwhile, Müller et al. (2012) showed that for the North Atlantic sea ice proxy IP25 decreased significantly in the early Holocene, while in the middle Holocene 7000-3000 years BC, sea ice gradually increased. The maximum value was gradually reached in late Holocene during 3000-300 years. For the Atlantic Arctic, the Chukchi Sea is an important sea area for sea ice drift to the North Atlantic, and its sea ice concentration was not always low during the early Holocene, while there were millennial oscillations and minimum values of sea ice in the Chukchi Sea during the Neoglacial (Anne et al., 2005).

Refence:

Müller J, Werner K, Stein R, et al. Holocene cooling culminates in sea ice oscillations in Fram Strait[J]. Quaternary Science Reviews, 2012, 47: 1-14.

De Vernal A, Hillaire-Marcel C, Darby D A. Variability of sea ice cover in the Chukchi Sea (western Arctic Ocean) during the Holocene[J]. Paleoceanography, 2005, 20(4).

Jennings A E, Knudsen K L, Hald M, et al. A mid-Holocene shift in Arctic sea-ice variability on the East Greenland Shelf[J]. The Holocene, 2002, 12(1): 49-58.

3. Result

3.1 Arctic temperature change

ll. 152 ff: How robust are the temperature changes? Are they statistically significantly different to internal changes. Therefore, applying a statistical test is helpful to estimate

the amount of internal variability between the two different periods, preferentially taking into account the serial correlations within the proxy-based estimations of temperature variability.

Reply: Thank you for your helpful comments and suggestions. We used box figure (see below) as well as t-tests to help estimate the amount of internal variability between the two different periods. Due to the small sample size of the Pacific Arctic reconstruction data, the temperature changes are only showing significance (p<0.10) on t-test.

[Figure]

ll. 172 ff: How significant are the changes between the Arctic and the Pacific region ? (i.e. -0.67 vs. +0.09.) Especially the Pacific trend seems to be statistically indistinguishable from a zero trend).

Reply: Thank you for your comments and suggestions. The t-test suggests the temperature changes in two region are significant (p<0.01). For winter temperature change it seems to be statistically indistinguishable from a zero trend, but for annual average or summer, there is a significant cooling.

ll. 191 ff: Also for the model-based differences of the sea ice a local statistical test on the spatial pattern including the effect of serial correlation is important to test the robustness and statistical significance of the according changes.

Reply: Thank you for your comments and suggestions. We modified the figure and perform spatial significance test for the figure of sea ice change.

ll. 202 ff: Changes in atmospheric circulation are also influenced to a high degree to internal variability – as such it is very important to use additional model simulations to back-up those changes, resulting from the CESM accelerated simulation. Moreover, why are the results of the orbital simulation are "more significant" than the one for the

all forcings? On Holocene time scales changes in orbital forcing on seasonal time scales exert a larger impact than the decadal-and sub-decadal changes caused by solar and volcanic activity. Therefore it is important to describe in greater detail how changes in solar and volcanic forcings are implemented into the accelerated CESM simulation.

Reply: Thank you for your comments and suggestions. As mentioned above, the difference between acceleration and non-acceleration simulation is the dampened and delayed response to external forcings in the deep ocean for the latter. There should be no big differences for the atmospheric circulation and surface climate response. We have added some results from other non-accelerated simulations to support our arguments (e.g. ECBilt-CLIO, IPSL and Trace-21k). Orbital forcing as the most obvious driver of long-term trend changes during the Holocene. Volcanic and TSI forcing have less impacts on long-term trends, and their role is more dominant on shorter timescales such as decadal and multi- decadal scale. However, the aim of our study is not to focus on these shorter timescales, so our analyses focus on orbital forcing and All forcing simulations.

3.3 EOF of SLP and UV wind regression and 3.4 The connection between Arctic dipole pattern and PDO

The whole sections lack a more thorough motivation on i) how the statistical concepts are used/defined and the ii) the robustness and statistical significance of the according regression patterns between the PCs and the underlying wind/sea ice fields. For instance, the PCs presented in Fig. 6 are (obviously) filtered with a low-pass filter. This should be accounted for when discussing and presenting the results.

Reply: We agree with this comment. We have clarified the objects of the statistical calculations as well as the definitions. We calculated the regression patterns between the UV wind/sea ice fields using the original time series of PC. And confirmed that the results passed the t-test. As for the PC in Figure 6, you are correct that it is filtered. As with the response to the comment on Figures below, we have revised the PC time series to a 50 model year sliding filter in the revision and clarified that in the figure caption.

Further, in addition to the UV regression, a Canonical correlation analysis would be better suited for this kind of investigation in section 3.3, since the rationale is to compare the common behavior of patterns (in this case the spatially resolved SLP and wind/sea ice fields.)

Reply: Thank you for your helpful suggestions. However, we believe that regression is a more suitable approach. We need not only to compare the common behavior, but more importantly we need to see the changes in sea ice and UV development based on sea level pressure between the early and late Holocene. Using only Canonical correlation analysis may see signals that are confounded by other factors and not really understand the effects of SLP on sea ice and UV.

A last point is again on the validity and model-dependence of the results based only on the accelerated simulation with CESM. This is in my opinion the weakest but most crucial point of the study.

Reply: Thank you for your important suggestions. As mentioned before, we have added some describing the validation of CESM simulations and analyze some results of unaccelerated simulations like TraCE-21ka, ECBilt-CLIO and IPSL, but the main focus will remain on the CESM results. The relevant description of the revised manuscript is in the introduction and discussion section.

4. Discussion

l. 291: Authors should formulate more nuanced that in this very version of the manuscript, results only apply to their few accelerated simulations with CESM that need to be compared with more recent, non-accelerated studies.

Reply: Thank you for your constructive comments and suggestions. We add a comparison with more recent, non-accelerated studies. The following description was added to the discussion section: "It can be assumed that our accelerated simulations can be consistent with the unaccelerated experiments in terms of the long-term climate evolution without involving changes in the deep ocean. We present some simple results to validate our results by comparing other non-accelerated simulations. We have selected three open-access unaccelerated simulations, TraCE-21ka, ECBilt-CLIO and IPSL. In summary, for the asymmetric cooling of SST, the simulations of ECBilt-CLIO as well as IPSL are similar to our results, while TraCE-21ka differs from our results. In particular, both ECBilt-CLIO and IPSL simulations show asymmetric cooling in the two Arctic regions and both have a greater cooling in the North Atlantic than in the North Pacific. The first transient simulations (Timm and Timmermann, 2007) based on the ECBilt-CLIO model with a horizontal resolution of about 5.6° × 5.5° and covering the past 21 ka, with the external forcing of ice cover, greenhouse gas concentration, and orbital configuration. The unaccelerated simulations using ECBilt-CLIO similarly reveal a regional asymmetry in temperature variability during the early-mid to late Holocene, with greater cooling in the North Atlantic (-0.26 °C cooling) than in the North Pacific (-0.02 °C cooling) (Supplementary Fig.3). The second nonaccelerated simulations based on IPSL ESM model (Braconnot et al., 2019) explored the relationship between climate change and vegetation over the past 6000 years. The IPSL ESM also captures the asymmetry characteristics of Arctic temperature variability between the mid Holocene (4-6ka BP) and late Holocene (0-2 ka BP),with -0,14 °C cooling in the Pacific Arctic and -0.18 °C cooling in the Atlantic Arctic (Supplementary Fig.3). The asymmetric changes are more pronounced if we focus only on the North Atlantic and North Pacific ocean regions. However, the results of TraCE-21ka are slightly different. TraCE-21ka is forced by the Earth's orbital parameters forcing, the greenhouse gas forcing, the meltwater flux forcing, and the continental ice sheets forcing. Its results show that the difference in annual mean temperature between the early-mid Holocene and late Holocene is significantly cooler in the Arctic and there is

a regional asymmetry in temperature changes between the two regions, with -0,94 °C cooling in the Pacific Arctic and -0.06 °C cooling in the Atlantic Arctic (Supplementary Fig.3). However, unlike NNU-Hol, the cooling in the Atlantic Arctic is smaller in magnitude than that in the Pacific Arctic in the asymmetric change of temperature in TraCE-21ka. This result might be attributed to the different external forcings of TraCE-21ka and NNU-Hol: there is still considerable residual ice-sheets and freshwater discharge in TraCE-21ka at about 8 ka BP which is not present in NNU-Hol."

l. 293: How should GHG changes, only changing very moderately in the pre-industrial period of the Holocene counteract any changes in orbital forcing ? If any, volcanic (and maybe in parts) negative periods of solar activity could counteract the negative trend in orbital forcing during the JJA season over the Arctic.

Reply:  Thank you for raising an important point here. We have revised this statement. We mean that the orbital simulation shows stronger asymmetric changes compare to the All forcing simulation. This implies that the combined effect of other forcings (solar irradiance, volcanic eruptions, greenhouse gases, and land use/land cover) and internal climate variability is offsetting this asymmetric changes (as opposite to the orital forcing). The contribution of different forcings is what we will need to study in the future. The reason why we mention "e.g. GHG" is that in future climate change, the GHG is an important factor that cannot be ignored.  We have revised this paragraph to make it more clearly  at the end of the Discussion section.

l. 284: The authors state that additional simulations should be used for investigations. Since those simulations are yet available authors should use them as an integral part of their revised study and thoroughly test their hypotheses with non-accelerated simulations and those carried out with different CMIP4-types of models.

Reply: Thank you for your valuable and insightful comments We have added a comparison with more recent, non-accelerated studies. See also reply for the Comment l.291.

Figures:

Fig 3.1: How does the Proxy (z-score) and the Model (°C) compare on the same axis ? In my opinion it would be necessary to show both on the same scale for an appropriate comparison.

Reply: Thank you for your comments. We've modified it. We still use the unit Celsius in the manuscript to describe the change in temperature.

[Figure]

Fig. 5: Please use units of hPa when presenting changes of sea level pressure fields.

Reply: Thank you for the nice reminder. We've modified it.

Fig 6, 9a and 10a: In this form of the presentation, the EOF pattern seem to carry normalized values (i.e. z-scores). In order to re-normalize the EOFs (i.e. eigenvectors), the patterns should be multiplied with the square root of their eigenvalue. Then the EOF patterns carry the units (in this case Pa(hPa) for SLP and K for SSTs, respectively). Eventually the according (original) PCs should be divided by the square root of the eigenvalue in order to show consistent patterns between EOFs and PCs. In addition, the temporal filtering should be indicated for the time series.

Reply: Thank you for your helpful comments. We have modified Fig. 6, Fig. 9, and Fig. 10 according to your method and labeled the units of the EOF modes. For the PC time series in Figure 6, we have adapted the original 20-200 year bandpass filtering to the current 50 model year running average filtering. The PC time series in Figure 6, Figure 9 show the results still after normalization.

Additional references / State-of-the art Holocene ESM simulations:

Transient Holocene simulation (6ka BP - 2ka BP) with interactive vegetation and phenology: https://vesg.ipsl.upmc.fr/thredds/catalog/work/p86mart/IPSLCM6/PROD/Holocene/TR6AV-Sr02/catalog.html

Braconnot, P., Zhu, D., Marti, O. and Servonnat, J.: Strengths and challenges for transient Mid- to Late Holocene simulations with dynamical vegetation, Clim. Past, 15(3), 997–1024, doi:10.5194/cp-15-997-2019, 2019

Braconnot, P., Marti, O., Crétat, J., Zhu, D., Sanogo, S., Balkanski, Y., Caubel, A., Cozic, A., Foujols, M.-A. and Servonnat, J.: Transient simulations of the last 6000 years with the IPSL model, in PMIP Workshop group P2FVAR., 2019.

Bader, J., Jungclaus, J., Krivova, N., Lorenz, S., Maycock, A., Raddatz, T., Schmidt, H., Toohey, M., Wu, C.-J. & Claussen, M., 2020: Global temperature modes shed light on the Holocene temperature conundrum. Nature Communications, 11: 4726. doi:10.1038/s41467-020-18478-6.

Dallmeyer, A., Claussen, M., Lorenz, S. J., Sigl, M., Toohey, M., and Herzschuh, U.: Holocene vegetation transitions and their climatic drivers in MPI-ESM1.2, Clim. Past Discuss. Clim. Past, 17, 2481–2513, https://doi.org/10.5194/cp-17-2481-2021, 2021.

New References:

Varma V, Prange M, Merkel U, et al. Holocene evolution of the Southern Hemisphere westerly winds in transient simulations with global climate models[J]. Climate of the Past, 2012, 8(2): 391-402.

Timm O, Timmerman A (2007) Simulation fo the last 21000 years using accelerated transient boundary conditions*. J Clim 20(17):4377–4401

Lu, Z., Liu, Z. Orbital modulation of ENSO seasonal phase locking. Clim Dyn 52, 4329–4350 (2019). https://doi.org/10.1007/s00382-018-4382-1

Jing Y, Liu J, Wan L. Comparison of climate responses to orbital forcing at different latitudes during the Holocene[J]. Quaternary International, 2022, 622: 65-76.

Holton J R, Haynes P H, McIntyre M E, et al. Stratosphere-troposphere exchange[J]. Reviews of geophysics, 1995, 33(4): 403-439.

GRIESER J, SCHONWIESE C D. Parameterization of spatio-temporal patterns of volcanic aerosol induced stratospheric optical depth and its climate radiative forcing[J]. Atmósfera, 1999, 12(2).

Malevich S B, Vetter L, Tierney J E. Global core top calibration of δ 18O in planktic foraminifera to sea surface temperature[J]. Paleoceanography and Paleoclimatology, 2019, 34(8): 1292-1315.

Tierney J E, Tingley M P. BAYSPLINE: A new calibration for the alkenone paleothermometer[J]. Paleoceanography and Paleoclimatology, 2018, 33(3): 281-301.

Tierney J E, Tingley M P. A Bayesian, spatially-varying calibration model for the TEX86 proxy[J]. Geochimica et Cosmochimica Acta, 2014, 127: 83-106.

Tierney J E, Malevich S B, Gray W, et al. Bayesian calibration of the Mg/Ca paleothermometer in planktic foraminifera[J]. Paleoceanography and Paleoclimatology, 2019, 34(12): 2005-2030.

---

## Author Response (AR2)

Review for the manuscript

Asymmetric changes of temperature in the Arctic during the Holocene

based on a transient run with the CESM

by Hongyue Zhang et al.

Submitted for publication in Climate of the Past
* * *
**General**

The revised version has been improved compared to the first submission and comments are mostly sufficiently addressed, especially in the context of adding and analyzing currently available Holocene transient simulations. Still, I am critical as to whether the impact of acceleration is as minor as the authors suggest, especially for long term changes on sea ice or other quantities. For example, Fig. 12 in the reply even shows a nice example that the acceleration technique has consequences on the evolution of patterns, such as the temperature profile.

Reply: Thank you for your recognition of the progress of our manuscript. We have carefully considered the comments and tried our best to address every one of them. We are concerned with Arctic temperature change, which primarily involves climate change processes on the land surface and sea surface at high latitudes. As expressed in the reference we exemplify, the accelerated model result has minor impact on the problem we are studying compared the unaccelerated model. Our conclusion is also supported by Figure 12 (Lu et al. 2019). Figure 12 shows the time series of the vertical temperature profiles. Y-axis is depth (m) and x-axis is accelerated year, and the contour is the anomalous temperature (°C). Black lines connect the maximum signals in the ocean. The acceleration causes longer adjustment times in the deep ocean compared to the surface ocean, leading to weakened and delayed responses in the deep ocean. However, our study is mainly concerned with the variation of the shallow ocean temperature. If we focus only on the ocean temperature in the range of 0-50 m, it can be seen from Fig. 12 that there is little difference in the comparison of temperature diffusion without and with acceleration by a factor of ten.

[Figure]

Fig. 12 Time sequence of the vertical temperature profile in a simple diffusion model under three acceleration scenarios: (upper panel) 100-fold acceleration, (middle panel) tenfold acceleration and (bottom panel) non-acceleration. (from Lu et al. 2019)

I also very much appreciate the additional work and investigations, especially related to the implementation of statistical tests. However, in the revised version the tests are poorly introduced and implemented and I encourage the authors to be more specific in the final version of the paper which statistical tests are applied and if they are appropriate for the respective purpose. For instance, the effect of serial correlation is still not addressed. In addition, it is important to select the level of confidence before the statistical test is carried out (typically alpha=0.05 for a two sided test) and eventually test if the numbers lie within or outside the confidence interval. For the trend test it is not mentioned at all which test has been applied (e.g. Mann-Kendall Test or any other Bootstrap method). To facilitate the description of statistical tests used, the authors should include a short caption into their Methods section and briefly describe which statistical tests are applied, including their specific setup and potential shortcomings (e.g. when the number of degrees of freedom is very small or the effect of serial correlation is not accounted for).

Reply: Thank you for your recognition of our work. The significance test used in the paper is mainly the students t-test, and the significance level is chosen as alpha=0.1 for two-tailed. In the revised manuscript, we have added a subsection in Section 2 specifying which statistical tests were applied as well as the EOF analysis method.

We have revised this description as:
"2.3 Analytical and Statistical Methods
We focus on long-term temperature changes in the Arctic during the Holocene. The significance test used in this study was calculated according to the two-tailed Students t-test at the 90% (alpha = 0.1) or 95% (alpha = 0.05) confidence level. The Students t-test was used to compare the means of two groups and determine if the difference in means is statistically significant and was also used to test the statistical significance of each grid in the figures below. The sample size of the Pacific Arctic region in the temperature proxy data is small and thus a small degree of freedom. We apply empirical orthogonal function (EOF) analysis, also known as principal component analysis (PCA), to sea level pressure changes in the Northern Hemisphere. EOF analysis is a standard analytical technique used in climate science to study patterns of spatial variability. EOF is obtained by computing the eigenvectors and eigenvalues of the spatially weighted covariance matrix of the temperature field. Applying EOF to the Northern Hemisphere sea level pressure is a common method to study the Arctic dipole mode. The objective is to show the variation of the Arctic dipole mode during different periods of the Holocene (0-2 ka BP, 5-8 ka BP). As described in Section 3 below, the second mod of EOF for the Holocene 5-8 ka BP period explains 11.5% and that for 0-2 ka BP period explains 16.3% of the sea level pressure variation."

In general, I think the manuscript should be published, but I still vote for "major revisions" until the statistical tests are carried out and described more thoroughly to back up the robustness of according results. For the re-revised version I list some minor comments below

Reply: Thank you for your comments. We have carefully considered the comments and tried our best to address every one of them.

**Specific**

Abstract:
The abstract is still in its original form. The authors should include the additional conclusions taking into account the new simulations and also mention the drawbacks and shortcoming of their study using the acceleration technique in the very beginning.

Reply: Thank you for pointing out our oversight. The new abstract was rewritten to add the shortcomings of the acceleration technique and the results of the verification of the Arctic temperature asymmetry using unaccelerated simulations. The results of the PDO section were removed in new abstract.

Introduction:
When presenting the additional simulations it would be also helpful to make the reader aware that the EC-Bilt model is an Earth System Model of Intermediate Complexity and therefore falls into another category of climate models. The comprehensive Earth System Model simulations with CCSM3 and IPSL are therefore better suited for a consistent comparison with the accelerated simulations the authors carry out with the CESM.

Reply: Thank you for your suggestion. We have added a description of EC-Bilt as an Earth system model of moderate complexity in the introduction section.

2.1 The CESM Model and the transient simulations
In the last paragraph the authors list studies covering transient Holocene simulations. In this list the more recent unaccelerated MPI-ESM (Bader et al., 2020) and IPSL (Braconnot et al., (2019) simulation are missing and should be added.

Reply: Thank you for your comments. We have added the information on non-accelerated models (IPSL and MPI-ESM) in the last paragraph of this section for the reader to have a more comprehensive understanding.

We have revised this description as:
"There are, to our knowledge, 5 sets of climate simulations published so far covering the entire Holocene period, namely ECBilt-CLIO [1], FOAM [2], TraCE-21ka [3], FAMOUS [4] and LOVECLIM [5]. Except for TraCE-21ka, these simulations are accelerated by different factors. Some unaccelerated simulations have also been published in recent years, such as those covering the Holocene 8ka BP and 6ka BP based on the MPI-ESM [6] and IPSL [7], respectively. The external forcings considered in these simulations are generally a part of the combination of the Orbital Forcing (ORB), the Greenhouse Gases (GHG), the continental ice sheets (ICE), the Meltwater Flux (MWF), the Volcanic Forcing, the Landuse Forcing and Ozone Forcing."

3.1 Arctic Temperature Change
Fig. 2 and following Figures: The temperature/sea ice/sea level pressure differences should be flagged with a symbol (e.g. hatching) for those areas being statistically significant different at the 5% level. Although the numbers mentioned in the Figure captions should contain an information on their statistical significance.

Reply: Thank you for your very helpful suggestions. To better illustrate the statistically significant areas on the figures, we performed a significance test using the Students t-test with a significance level of alpha=0.1 for two-tailed. The regions that passed the significance test under this calculation are marked with a dotted conformation. As shown in Figure 2 below, we modified Figures 2,4,5,6,7.

[Figure]

3.2 Sea Ice Change (Aice, March) and SLP Change

ll. 439 new version: the authors mention some physical mechanisms linking the change in radiation to albedo and heat storage changes, quoting Dai, (2021). The study of Dai (2021) used however a profoundly different setup (500 year CO2-surface albedo vs. 2000 year control) with CESM 1.0. My suggestion is to use the results based on the accelerated CESM simulations the authors carried out, if mechanisms are investigated and interpreted in the context of Holocene climate change.

Reply: Thank you for your valuable advice. We want to emphasize the enhancement of positive feedback related to changes in sea ice, and the enhancement of temperature asymmetry caused by radiation to heat changes. You are correct, Dai et al. (2021) involves research on heat storage and CO2-surface albedo. Due to the complexity of thermal storage and other forcing radiative effects and the fact that they are not the primary objective of our consideration, we did not perform calculations for CESM data. We have modified the description in 439.

ll. 488 new version: As stated in the previous comments, I still have concerns with the formulation "The results of the ORB simulations are more significant than those of the AF, [ … ]". Conceptually, a result can be statistically different to a reference/control on a certain level of confidence or not. Maybe authors can clarify this statement whether

this is linked to a statistical statement or in the context of a comparison that one simulations shows larger/lower values compared to another one.

Reply: Thank you for your valuable comments. In the new revised manuscript we clarified that the ORB simulations compared to AF simulations are values comparisons. We modify the statement that ORB is more "significant" in the manuscript because we did not perform a significance test for the difference between the two Arctic regions for two simulations.
We revise line 488 to read as follows "Compared with the AF results, the ORB simulation show that the sea level pressure changes in the two regions are more contrasting and the difference is larger. This suggests that orbital forcing plays a contributing role in generating this asymmetry than other forcings."

Fig. 5: I assume the numbers at the color bars are still in units of Pa, although the labeling has changed to hPa. Authors should also change the numbers displayed at the colorbar for consistency.

Reply: Thank you for pointing that out. We have revised Figure 5.

3.3 EOF of SLP and UV wind regression
Unfortunately, even authors state in their reply that the motivation for using EOF/regression was implemented in the revised version, it is still not presented. For instance, hemispheric EOFs can be misleading in their pattern structure (cf. Discussion Ambaum, 2001 on the physical plausibility of the Arctic Oscillation concept (1 EOF SLP Northern Hemisphere). Besides that, authors should explain why EOF analysis presents a meaningful tool, even on hemispheric scales for investigating SLP changes (e.g. because of potential representation of real teleconnection patterns) and why the according PCs could be used as a basis for regression. It should be tested or at least mentioned if this teleconnection structure represented by the EOF is actually real or just an artefact of the Eigenanalysis. In the present form results are just presented without introducing any general background shortcomings of the methods.

Reply: Thank you for mentioning this point. In the manuscript above, we pointed out the asymmetric variability characteristic of Arctic temperature and the similar asymmetry of sea ice and sea level pressure variability during the Holocene. Therefore, we would like to further illustrate the connection between them. Thus we explore the physical processes between them by analyzing the Arctic dipole. The EOF method is commonly used to study the Arctic dipole in the past studies (Wu et al., 2006; Wang et al., 2000; Skeie et al., 2000). This is exactly our motivation for using EOF. Our EOF for the SLP in the manuscript is not on the hemispheric scale. The first leading mode of the EOF corresponds to the Arctic Oscillation pattern, and the second mode that corresponds to the Arctic dipole. So in the manuscript we focus on the second mode. Regression of the PC time series shows how the corresponding temperature, sea ice, and wind change under the action of the Arctic dipole, thus helping us to understand

the physical processes involved.

l. 550: A hypothesis can not be verified. It can only be falsified, whether it is consistent or not with a null hypothesis on a certain level of confidence. In addition, similar to previous chapters, all numbers mentioned in the text and figure captions need to be tested on their statistical significance.

Reply: Thank you for your comments. The hypothesis we describe in line 550 of the manuscript is not a statistically significant hypothesis. Rather, it refers to a conjecture we made in line 292 about the contribution of sea ice and sea level pressure changes to temperature asymmetry. We have revised this "hypothesis" in the manuscript. We have also added a statistical test description to this paragraph.

3.4 The connection between Arctic Dipole pattern and PDO
I suggest to completely leave out the entire section. The PDO is never mentioned in the introduction or elsewhere and the analysis does not add anything to the conclusions presented in the manuscript. There is still no motivation given for the analysis (Why separate positive and negative PDO years and perform an EOF ?!?). In addition, the interpretation of according results is more than speculative.

Reply: Thank you for your important suggestion. We removed this section as you suggested. Our ideas in this section demonstrate that different phases of the PDO in the Holocene are associated with Arctic dipole modes. We separate PDO in different phases and perform EOF to explore the relationship between PDO phase and the Arctic Dipole. We conclude that the positive phase of the PDO is dominant in the early-mid Holocene and the Arctic Dipole is weak. In the Late Holocene, the PDO negative phase is dominant, and the Arctic dipole is stronger. We argue that The potential phase of the PDO dominates the SLP to form the Arctic dipole mode. The PDO is also closely related to the orbital forcing. Therefore, it is speculated that the ORB orbital forcing affects the Arctic dipole by modulating the PDO phase, and then contributes to the asymmetry of the Arctic temperature.

4 Discussion
1st paragraph, l. 699 new version: "are responsible" – I suggest to reformulate to "are an important factor"
Reply: Thank you for your suggestion. We've modified it.

l 777: please re-formulate "new insight" to "additional insights on Holocene time scales" – The study is not the first one addressing climateteleconnections in the Arctic realm.
Reply: Thank you for your comments. We've modified it.

l. 780: please remove the sentence "These results can have useful implications on predicting…" - The study has no dedicated chapter on predictability of the Arctic climate. The basic processes authors motivate have been already published. (e.g. the importance of sea ice and atmospheric circulation on regional Arctic processes).

Reply: Thank you for your helpful comments. We've modified it.

Supplementary:
Fig. S3. Please also carry out according statistical significance tests for the plots and the numbers mentioned in the Figure caption.

Reply: Thank you for your suggestion. We've modified it.

---

## Author Response (AR3)

Review for the manuscript

Asymmetric changes of temperature in the Arctic during the Holocene

based on a transient run with the CESM

by Hongyue Zhang et al.

Submitted for publication in Climate of the Past
* * *
Thank you for your suggestions on our manuscript. We do our best to address each of them.

Line 27. Replace '…verify…' by '… confirm the occurrence of the …'
Reply: Thank you for your suggestion. Fixed.

Line 28. Replace '…using unaccelerated simulations of ECBilt-CLIO, IPSL, and Trace21k.' by '… in unaccelerated simulations using ECBilt-CLIO, IPSL, and in Trace21k.
Reply: Thank you for your comments. Fixed.

Line 163. The structure of the paragraph is not clear for me. You first state line 159 that only 5 simulations have been published and then line 163 that they are two more that are also published. Why not stating immediately that they are 7 that are published?
Reply: Thank you for pointing that out. Line 159 declares five simulations that completely cover the Holocene period (0-11700 ka BP), while the other two simulations declared in line 163 cover 6 ka and 8 ka respectively but not the entire Holocene period. We have modified the sentences as "Additional unaccelerated simulations, such as the simulations based on MPI-ESM (Bader et al., 2020) and IPSL (Braconnot et al., 2019) covering only part of the Holocene period (0-8 ka BP and 0-6 ka BP, respectively), have also been published in recent years."

Line 204 'for each grid point' instead of 'of each grid'
Reply: Thank you for your suggestion. Fixed.

Line 205. You mention a small number of degree of freedom. One reviewer mentions several times the problems potentially related to serial correlation. If I understand well, this has not been taken into account. You must then mention at least in the methodology

that 'The potential impact of temporal and spatial correlation is not taken into account in the analyses'.

Reply: Thank you for pointing out our oversight. We have added this description in the methods section.

Line 309. Replace 'This suggests that orbital forcing plays a contributing role in generating this asymmetry more than other forcings' by 'This suggests that orbital forcing plays a dominant role in generating this asymmetry.

Reply: Thank you for your suggestion. We've modified it.

Line 337. Please explain how you determined that 'The first and second leading modes are significantly separated at 95% confidence level.'

Reply: Thank you for the comment. What we want to express here is that the first and second leading modes of the EOF are statistically independent of each other. So we can use the second mode of the EOF in SLP to represent the pattern of Arctic Dipole. We have modified 'significantly separated' to 'statistically independent'.

Line 504. Change 'Using the non-accelerated model to verify the Arctic temperature asymmetry' by 'Using no-accelerated simulations to test the robustness of our results the Arctic temperature asymmetry, we found that.

Reply: Thank you for your suggestion. We've modified it.

---

## Author Response (AR4)

Review for the manuscript

Asymmetric changes of temperature in the Arctic during the Holocene

based on a transient run with the CESM

by Hongyue Zhang et al.

Submitted for publication in Climate of the Past
* * *
Thank you for deciding to accept our article. We have revised the technical corrections.

Line 295: 'The first and second leading modes are statistically independent at 95% confidence level.' You should explain the methodology used to determine that the two models are independent or give a reference.

Reply: Thank you for your suggestion. We used the equation 24 from North et al.(1982) as the statistical test. We modified the description of 2.3 in Method and data and Line 295 by the following statement.

In 2.3: "Equation 24 from North et al., (1982) was used as a statistical test to evaluate the separation of EOF eigenvalues (leading modes)."

Line 295: "The first and second leading modes are statistically independent at 95% confidence level on North significance test (North et al., 1982)."